## PROCEEDINGS A

statistics, differential equations

mathematical modelling, equation learning, uncertainty quantification

**Author for correspondence:**
Simon Martina-Perez
e-mail: martinaperez@maths.ox.ac.uk

# Bayesian uncertainty quantification for data-driven equation learning

Simon Martina-Perez[1], Matthew J. Simpson[2] and Ruth E. Baker[1]

[1]Mathematical Institute, University of Oxford, Oxford, UK
[2]School of Mathematical Sciences, Queensland University of Technology, Brisbane, Australia

SM-P, 0000-0001-8596-8595; MJS, 0000-0001-6254-313X; REB, 0000-0002-6304-9333

Equation learning aims to infer differential equation models from data. While a number of studies have shown that differential equation models can be successfully identified when the data are sufficiently detailed and corrupted with relatively small amounts of noise, the relationship between observation noise and uncertainty in the learned differential equation models remains unexplored. We demonstrate that for noisy datasets there exists great variation in both the structure of the learned differential equation models and their parameter values. We explore how to exploit multiple datasets to quantify uncertainty in the learned models, and at the same time draw mechanistic conclusions about the target differential equations. We showcase our results using simulation data from a relatively straightforward agent-based model (ABM) which has a well-characterized partial differential equation description that provides highly accurate predictions of averaged ABM behaviours in relevant regions of parameter space. Our approach combines equation learning methods with Bayesian inference approaches so that a quantification of uncertainty can be given by the posterior parameter distribution of the learned model.

## 1. Introduction

Many phenomena in Nature arise as a result of complex interactions between individual agents at the microscale that give rise to emergent properties at the macroscale.

Understanding the mechanistic basis for the observed macroscale behaviour, in order to gain fundamental insights into biological phenomena, is one of the key challenges in biology.

Mathematical models are well placed to help provide such insights, providing a rigorous framework where hypotheses can be generated, tested and refined. While interactions between individual agents can be naturally described by agent-based models (ABMs) that prescribe precise rules for the interactions between agents [1–4], predicting the macroscale behaviour of ABMs can be a challenging task, since their governing equations are often intractable and stochastic simulations can be computationally expensive, often prohibitively so in the context of parameter sensitivity analysis or parameter inference [5–9]. This makes differential equation models an indispensable tool to describe the expected macroscale properties of the population. The benefits of differential equation models include the fact that they are relatively fast to solve numerically, their different terms often carry a physical interpretation and they can be explored using a range of analytical and numerical approaches. Understanding how such a model can be parametrized, then, can provide key insights into the system under consideration, and aid in making quantitative as well as qualitative predictions [10].

Traditional approaches to mathematical modelling use experimentally derived mechanistic hypotheses to derive differential equation models in which the various terms of a given model are designed to describe the hypothesized mechanisms for that scenario. Calibration of the model to data then involves finding the parameters that minimize the discrepancy between the model output and data. The ensuing, iterative process of testing and refining the model against further experimental data allows the original hypotheses to be refined, and so new insights gained.

Equation learning (EQL) methods take a different approach to model building, aiming to infer the dynamical systems model that best describes given time-series data by leveraging statistical and machine learning tools to learn the appropriate terms of a differential equation model directly from the data. In particular, the PDE-FIND algorithm [11,12] takes as input quantitative data, together with a library of candidate terms for a partial differential equation (PDE) model, and aims to learn which terms to include in the PDE model, as well as their coefficients. Algorithm hyperparameters can be tuned to enable a balance between the requirement for a good model fit and the desire for a simple, interpretable model.

EQL methods have rapidly gained popularity, mainly owing to increases in computational power, and a number of other techniques to establish models from data now exist. For example, biologically informed neural networks [13], an extension of physically informed neural networks [14], have been developed to learn the different terms of a PDE model without the need to specify a library of possible terms. Furthermore, a major advance has come from the use of techniques such artificial neural networks [15] to accurately recover models from artificially generated noisy data from PDEs.

The fact that EQL can discover previously undetected mechanisms, discriminate between competing models or estimate biological quantities of interest that are difficult to measure experimentally makes EQL attractive to scientists working with real-world data. However, practitioners wishing to develop models that they can use in real-world settings require, in addition, a thorough quantification of uncertainty [10,16–19]. This need comes from the fact that the, often significant, noise in real-world data can impact the models predicted by EQL methods, and hence the predictive capability of the models for unseen data or scenarios [20,21]. For example, Nardini *et al.* [8] have recently shown, through the use of several case studies, that it is possible to infer differential equation models that describe noisy data generated by stochastic ABMs. However, the stochasticity in the ABM results in variability in the learned macroscale differential equation. This means that, for a particular realization generated from a stochastic model, the learned differential equation is a point estimate of the underlying differential equation, and there is no quantification of uncertainty in the learned equation.

Recently, some authors [12,22] have begun to address this problem by analysing the robustness of PDE-FIND with noisy or sparse data. For example, Rudy *et al.* [12] investigate how the learned PDE varies as the numerical solution of a ground truth PDE is corrupted by additive noise, while Li *et al.* [22] investigate how to increase the signal-to-noise ratio of a dataset prior to the use of

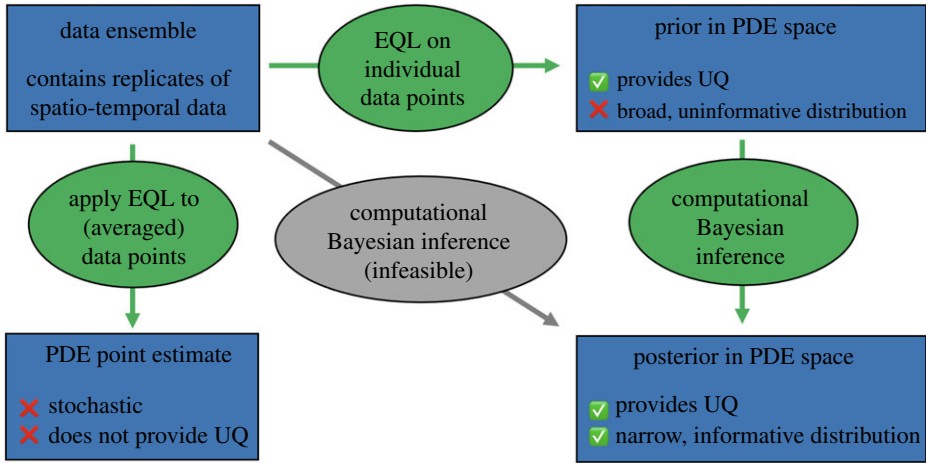

**Figure 1.** Proposed framework for uncertainty quantification (UQ) in equation learning (EQL). An ensemble of noisy datasets are used to create an informative prior distribution, which can then be used to obtain a posterior parameter distribution for the learned PDE. (Online version in colour.)

EQL techniques. While both works show that model parameters can be retrieved to within an impressive margin of error when the data are corrupted with relatively small amounts of noise, these approaches give no statistical quantification of the uncertainty in model predictions, nor do they address how to deal with significant noise levels.

In this work, we demonstrate that noise can significantly impact both the structure and the values of the parameters of learned differential equation models, rendering uncertainty quantification a crucial component of the EQL process. As such, our overarching aim is to develop and showcase a method for uncertainty quantification in the context of EQL, where we harness the immense computational efficiencies of PDE-FIND in learning point estimates of governing equations, together with the power of computational Bayesian inference in evaluating the level uncertainty in the learned equation. Figure 1 shows our proposed framework for uncertainty quantification. We start from the basis that it is possible to collect an ensemble of spatio-temporal datasets from a given system, and develop an approach to understand how the data can be used to learn a governing equation while simultaneously estimating the uncertainty in that learned equation.

Our motivation is thus: on the one hand, PDE-FIND provides a computationally cheap method to obtain a point estimate for the governing equation from a single time series. However, when the data are noisy, the individual predictions are unreliable. This is seen on the left-hand side of figure 1, where this approach is identified by obtaining a point estimate of the PDE. On the other hand, the field of computational Bayesian inference provides a number of methods to estimate, for a given model and data, posterior parameter distributions, i.e. it provides estimates of model parameters and quantifies the uncertainty in those estimates. In principle, computational Bayesian inference approaches could be used directly with the candidate library of the PDE-FIND method to estimate the posterior distribution of the library coefficients. However, owing to the very large number of candidate terms in the PDE-FIND library, the computational cost associated with applying methods for computational Bayesian inference on the entire high-dimensional parameter space is generally prohibitive. Instead, we propose a framework that combines the strengths of each approach: first, we train the PDE-FIND algorithm on individual datasets from the ensemble to obtain an informed prior parameter distribution (top row of figure 1). While this prior distribution will propably be relatively broad and uninformative of the uncertainty in the model, it can still be used to vastly reduce the dimensionality of the inference problem. Such a reduction in dimensionality makes it feasible to find an informative posterior distribution using computational Bayesian approaches (right-most column of figure 1).

In this work, we demonstrate the potential of this approach using synthetic data generated from a widely used ABM that describes the behaviour of a motile and proliferative cell population and can be coarse-grained to a mean-field PDE that accurately describes ABM dynamics in certain regions of parameter space. In §2, we describe the ABM and discuss in detail its relation with a governing PDE, as well as the PDE-FIND algorithm. In §3, we demonstrate both that the PDE models learnt using PDE-FIND are intrinsically variable when the data are noisy and that PDE-FIND can learn unphysical models. We provide an explanation for this in terms of the objective function of the PDE-FIND algorithm. In §4, we propose a method to combine methods for Bayesian inference with PDE-FIND in order to learn the structure of the governing PDE model, construct a prior parameter distribution for the PDE model and infer the posterior parameter distribution of the learned PDE model. We conclude in §5 with a discussion of our results, and avenues for future research. Code for all algorithms can be found at https://github.com/simonmape/UQ-for-pdefind.

## 2. Models and equation learning methodology

We begin by describing the ABM and briefly outlining how to derive the corresponding coarse-grained PDE model, and then we provide details of the PDE-FIND algorithm.

### (a) Agent-based model

ABMs allow practitioners to investigate the collective behaviour of a population of individuals based on a description of the behaviour of individuals within that population. Here, in order to take into account the interactions between individuals of the population, we follow the volume-exclusion model presented in [4,23–26] for a population of agents that move and proliferate according to a discrete random walk model. This is a simple model that can be used to analyse a range of phenomena, including the collective migration of cells in a tissue, for example.

We assume that agents occupy sites on a square lattice of spacing $\Delta$, so that their possible locations are $(i\Delta, j\Delta)$, where $(i, j)$ are integer coordinates, and volume exclusion entails that at most one agent can occupy a lattice site at any given time. We have $1 \leq i \leq I$ and $1 \leq j \leq J$, and throughout this work we take $I = 200$ and $J = 20$. A pseudo-one-dimensional initial condition is taken by initially populating all lattice sites with $90 \leq i \leq 110$ and leaving the rest of the lattice empty. We impose zero flux boundary conditions at all boundaries so that agents cannot leave the lattice and use time step $\tau$ to advance the simulations through time, with $T = 1000$ time steps in total for each simulation. Let $N(t)$ denote the number of agents on the lattice at time $t$. The parameter $p_m \in [0, 1]$ specifies the attempted movement probability of each agent in a time interval of duration $\tau$, and $\rho \in [-1, 1]$ the left–right bias in movements. Similarly, the parameter $p_p \in [0, 1]$ specifies the attempted proliferation probability of each agent in a time interval of duration $\tau$.

At each time step, $\tau$, a random sequential updating procedure is carried out: $N(t)$ agents are selected, one at a time, with replacement, and are allowed to attempt a movement or proliferation event. When an agent is selected, $S_1 \sim U(0, 1)$ is drawn. If $S_1 \leq p_p$ then the agent attempts to proliferate by placing a daughter agent into one of the randomly chosen nearest-neighbour sites. If the target site is occupied, then the proliferation event is aborted. If $p_p < S_1 \leq p_p + p_m$ then the agent attempts to move to one of its nearest-neighbour lattice sites. A second random number $S_2 \sim U(0, 1)$ is drawn and the target site is chosen according to the rules in table 1. As for proliferation, if the target site is occupied then the movement event is aborted. If $S_1 > p_p + p_m$ then the agent does not attempt to move or proliferate. For convenience, we take $\Delta = 1$, $\tau = 1$ and $p_m = 1$ and consider the effects of varying $p_p$ and $\rho$.

Let $C_{ij}^k(t)$ denote the occupancy of site $(i, j)$ at time $t$ in simulation $k$, so that $C_{ij}^k(t) = 1$ if $(i, j)$ is occupied by an agent at time $t$ and $C_{ij}^k(t) = 0$ if it is empty. We can average the site occupancy over

**Table 1.** Algorithm by which an agent at site $(i, j)$ selects a target site to move into.

| move chosen | target site | probability | where random number $S_2$ falls |
|---|---|---|---|
| vertically up | $(i, j - 1)$ | $\frac{1}{4}$ | $0 \leq S_2 \leq \frac{1}{4}$ |
| vertically down | $(i, j + 1)$ | $\frac{1}{4}$ | $\frac{1}{4} \leq S_2 \leq \frac{1}{2}$ |
| horizontally left | $(i - 1, j)$ | $\frac{1-\rho}{4}$ | $\frac{1}{2} \leq S_2 \leq \frac{1}{2} + \frac{1-\rho}{4}$ |
| horizontally right | $(i + 1, j)$ | $\frac{1+\rho}{4}$ | $\frac{1}{2} + \frac{1-\rho}{4} \leq S_2 \leq 1$ |

the columns of the lattice, defining the mean occupancy of column $i$ at time $t$ in simulation $k$ as, for $1 \leq i \leq I$,

$$C_i^k(t) = \frac{1}{J} \sum_{j=1}^{J} C_{ij}^k(t) \tag{2.1}$$

to give a one-dimensional averaged agent density profile for simulation $k$.

### (i) Coarse-grained PDE model

To make progress in deriving a coarse-grained PDE equivalent, we first note that the choice of a pseudo-one-dimensional initial condition means that we can consider deriving a one-dimensional PDE for $c(x, t)$, the density of agents at position $x$ at time $t$, without making explicit reference to the spatial coordinate $y$. We use $\langle C_i(t) \rangle$ to denote the average probability of occupancy of lattice site $i$ at time $t$, for $1 \leq i \leq I$, where the average is taken over $K$ simulations,

$$\langle C_i(t) \rangle = \frac{1}{K} \sum_{k=1}^{K} C_i^k(t) = \frac{1}{J \cdot K} \sum_{k=1}^{K} \sum_{j=1}^{J} C_{ij}^k(t). \tag{2.2}$$

We now consider the change in average occupancy of site $i$ over a time step of duration $\tau$ to write

$$\langle C_i(t+\tau) \rangle - \langle C_i(t) \rangle = \frac{(1+\rho)}{4} p_m \langle C_{i-1}(t) \rangle \left(1 - \langle C_i(t) \rangle \right) + \frac{(1-\rho)}{4} p_m \langle C_{i+1}(t) \rangle (1 - C_i(t))$$
$$- \frac{(1+\rho)}{4} p_m \langle C_i(t) \rangle \left(1 - \langle C_{i+1}(t) \rangle \right) - \frac{(1-\rho)}{4} p_m \langle C_i(t) \rangle \left(1 - \langle C_{i-1}(t) \rangle \right)$$
$$+ \frac{1}{2} p_p \langle C_{i-1}(t) \rangle \left(1 - \langle C_i(t) \rangle \right) + \frac{1}{2} p_p \langle C_{i+1}(t) \rangle (t) \left(1 - \langle C_i(t) \rangle \right), \tag{2.3}$$

where the first four terms on the right-hand side correspond to changes in occupancy owing to agent movement and the final two to agent proliferation. Note that, in writing down this conservation statement, we have implicitly assumed that lattice site occupancies are independent, so that, for example, the average probability that site $i$ is occupied and site $i \pm 1$ is unoccupied can be written as $\langle C_i(t) \rangle (1 - \langle C_{i\pm1}(t) \rangle)$. This is a standard assumption called the mean-field approximation [27–29].

We then identify $\langle C_i(t) \rangle$ with the continuous density $c(x, t)$, Taylor expand the resulting equation and take limits as $\Delta, \tau \to 0$, to arrive at the following PDE:

$$c_t = D c_{xx} - V[c(1 - c)]_x + P[c(1 - c)], \tag{2.4}$$

where

$$D = \lim_{\Delta, \tau \to 0} \frac{p_m \Delta^2}{4\tau}, \quad V = \lim_{\Delta, \tau \to 0} \frac{p_m \Delta \rho}{2\tau} \quad \text{and} \quad P = \lim_{\tau \to 0} \frac{p_p}{\tau} \tag{2.5}$$

and the subscripts $x$ and $t$ denote partial derivatives. For the full derivation and details, we refer the reader to [24–26].

Identification of the ABM with a coarse-grained macroscale PDE model motivates us to investigate the performance of EQL methods trained on data generated by the ABM, since

**Table 2.** The mean-field PDE models describing evolution of the mean population density over time for the three example cases used in this work.

| case | $p_m$ | $\rho$ | $p_p$ | coarse-grained PDE |
|---|---|---|---|---|
| I: no bias, no proliferation | 1.0 | 0.0 | 0.0 | $c_t = 0.25c_{xx}$ |
| II: bias, no proliferation | 1.0 | 0.075 | 0.0 | $c_t = 0.25c_{xx} - 0.0375[c(1-c)]_x$ |
| III: proliferation, no bias | 1.0 | 0.0 | 0.01 | $c_t = 0.25c_{xx} + 0.01[c(1-c)]$ |

the PDE accurately describes the time evolution of the expected value of the density profile and so we can evaluate the performance of EQL methods against equation (2.4). Note that, in order for equation (2.4) to provide an accurate description of the averaged dynamics of the ABM, we require that the assumption of lattice-site occupancy independence (i.e. the mean-field assumption) approximately holds. Typically, this requires $p_p$ and $|\rho|$ to be small relative to $p_m$ [23,27,30].

### (ii) Comparison of the ABM and PDE model predictions

As test cases for learning the governing equations from data, we explore three different parameter regimes in the model, which each correspond to a biologically relevant setting: in case I, we consider agents moving without bias and without proliferation ($\rho = p_p = 0$); in case II, we consider agents moving with bias but without proliferation ($\rho = 0.075$ and $p_p = 0$); and in case III, we consider agents moving without bias but with proliferation ($\rho = 0$ and $p_p = 0.001$). Table 2 outlines these different cases, along with a statement of the corresponding coarse-grained PDE model.

Figure 2 shows results from simulation of the ABM, averaged over different numbers of realizations, alongside the solution of the corresponding PDE model. The PDE model is solved numerically using the PyPDE package [31], which solves the PDE using the method of lines by discretizing space using the grid on which spatial data for the ABM have been collected. The resulting ordinary differential equations (ODEs) are solved using a fourth-order Runge–Kutta method on the domain $x \in [0, 200]$ with space discretization $\Delta x = 10^{-3}$, and constant time discretization $\Delta t = 10^{-4}$.

We make two observations. First, we note that the solutions of the PDE models accurately predict the dynamics of the ABM in the chosen parameter regimes (see also electronic supplementary material, figure S1), so that we have a 'ground truth' PDE against which to benchmark the EQL methodology. Second, at early times the density profiles are very similar for the three different cases, but at later times differences due to the effects of bias and proliferation are clearly discernible. This observation implies that the EQL methodology will require data on sufficiently long time scales to be able to accurately learn the correct PDE model.

### (b) Equation learning: PDE-FIND

In the following, assume that we have time-series data for an unknown function $u(x,t)$ on a grid of $n$ points in time and $m$ points in space. These data are stored in a matrix $U \in \mathbb{R}^{n \times m}$. We assume that the data are a noisy discretization of a function $u(x,t)$, the solution of an unknown PDE, and the aim is to learn the PDE that best describes the governing equation of the observed data. Henceforth, and to avoid confusion, we will write $U(x,t)$ for the observed data, $u(x,t)$ for the learned PDE and $c(x,t)$ for the solution to the coarse-grained PDE defined in equation (2.4). We follow Rudy *et al.* [12] in assuming that the PDE governing $u(x,t)$ is given in the following form:

$$u_t = \mathcal{N}(u, u_x, u_{xx}, \ldots), \qquad (2.6)$$

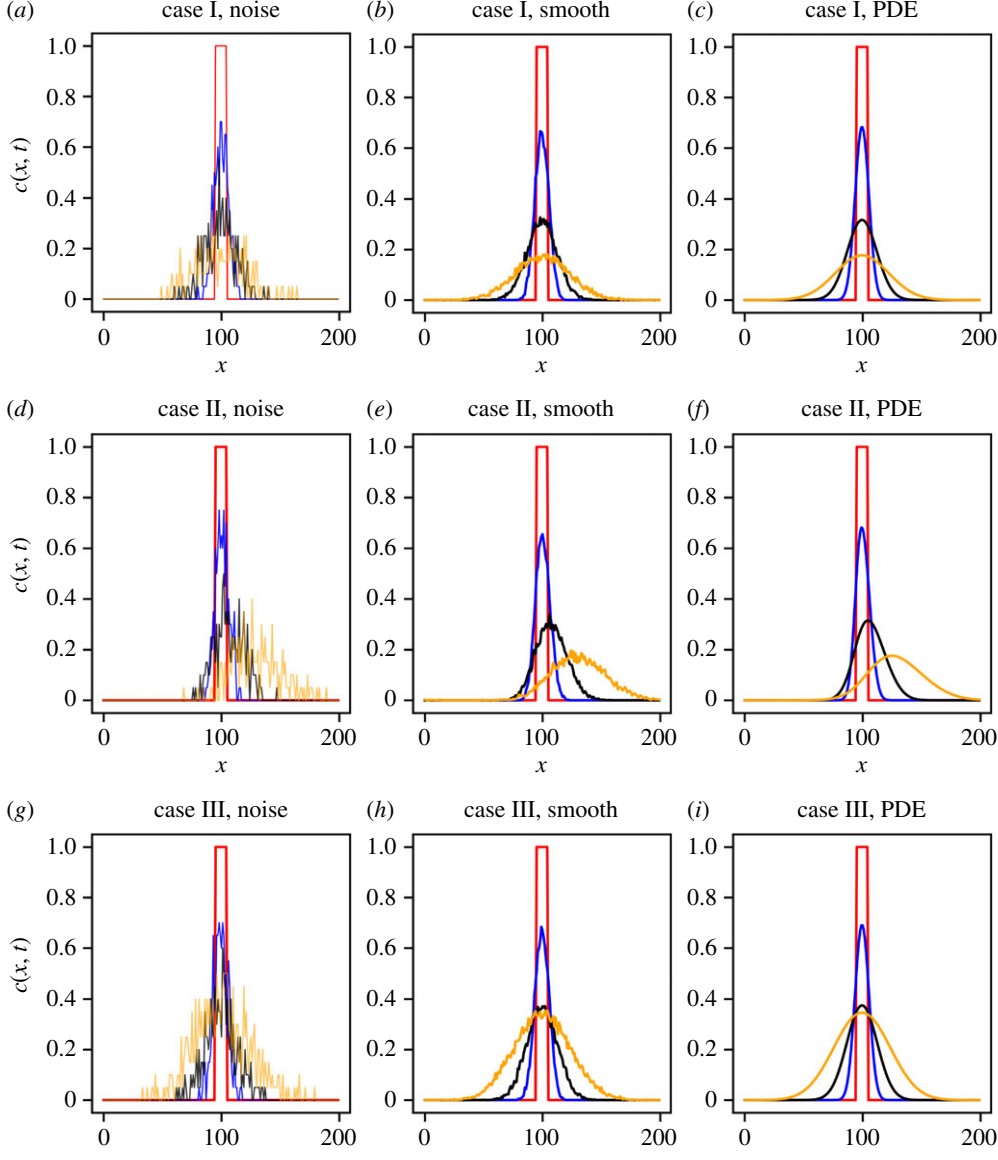

**Figure 2.** Typical one-dimensional density profiles. Plots (a,d,g) are from a single realization of the model ($K = 1$), plots (b,e,h) are averaged over 50 simulations ($K = 50$) and plots (c,f,i) are the solutions of the corresponding coarse-grained PDE. In each plot, we show the density at times $t = 0, 50, 150, 500$ (red, blue, black, orange). (Online version in colour.)

where $\mathcal{N}$ is a nonlinear function of $u(x,t)$ and its partial derivatives. Furthermore, it is assumed that $\mathcal{N}$ is a linear combination of a finite number of distinct library terms so that we can write

$$u_t = \sum_{i=1}^{N_\ell} \mathcal{N}_i(u, u_x, u_{xx}, \ldots)\, \xi_i, \tag{2.7}$$

for coefficients $\xi_i$, where $i = 1, \ldots, N_\ell$. By design, we will specify that $\mathcal{N}$ has polynomial nonlinearities, as is common in many equations in the natural sciences, and we note that equation (2.4) falls within this class of PDEs. The aim of PDE-FIND is then to select, from the large library of terms $\mathcal{N}_i$, for $i = 1, \ldots, N_\ell$, a small subset of relevant terms.

The first step of the PDE-FIND pipeline is to numerically approximate both sides of equation (2.7). This is done by estimating derivatives of the data with respect to space and time. The standard PDE-FIND implementation in [12] takes finite difference approximations when the data contain little noise and polynomial differentiation when the data are very noisy. The data and their derivatives are combined in a matrix $\Theta(U)$, where each column of $\Theta$ contains all of the values of a particular candidate function across the entire $n \times m$ grid. For example, if the candidate library consists of all polynomials up to degree 2 and non-mixed derivatives up to second order, $N_\ell = 9$ and $\Theta(U)$ will look like

$$\Theta(U) = [1, U, U^2, U_x, UU_x, U^2U_x, U_{xx}, UU_{xx}, U^2U_{xx}].  \tag{2.8}$$

As a result, if there are $N_\ell$ terms in the candidate library, $\Theta(U)$ is an $n \times m \times N_\ell$ matrix.

The left-hand side of equation (2.7) is similarly approximated, and we obtain a linear matrix equation representing the PDE evaluated at the data points,

$$U_t = \Theta(U)\boldsymbol{\xi},  \tag{2.9}$$

where $\boldsymbol{\xi} = [\xi_1, \ldots, \xi_{N_\ell}]^T$. Taking the same example for $\Theta(U)$ as in equation (2.8), this matrix equation is of the form

$$
\begin{pmatrix} U_t(x_0, t_0) \\ U_t(x_1, t_0) \\ U_t(x_2, t_0) \\ \vdots \\ U_t(x_{n-1}, t_m) \\ U_t(x_n, t_m) \end{pmatrix} = \begin{pmatrix} 1 & U(x_0, t_0) & U^2(x_0, t_0) & \ldots & U^2U_{xx}(x_0, t_0) \\ 1 & U(x_1, t_0) & U^2(x_1, t_0) & \ldots & U^2U_{xx}(x_1, t_0) \\ 1 & U(x_2, t_0) & U^2(x_2, t_0) & \ldots & U^2U_{xx}(x_2, t_0) \\ \vdots & \vdots & \vdots & \ddots & \vdots \\ 1 & U(x_{n-1}, t_m) & U^2(x_{n-1}, t_m) & \ldots & U^2U_{xx}(x_{n-1}, t_m) \\ 1 & U(x_n, t_m) & U^2(x_n, t_m) & \ldots & U^2U_{xx}(x_n, t_m) \end{pmatrix} \begin{pmatrix} \xi_1 \\ \xi_2 \\ \xi_3 \\ \vdots \\ \xi_8 \\ \xi_9 \end{pmatrix}.  \tag{2.10}
$$

Note how this representation shows that each row in the matrix equation represents the governing dynamics behind the data at one point in time and space. The values of the coefficients $\xi_i$ determine the form of the PDE, and so the aim is to learn the coefficients $\xi_i$ in some sense 'optimally'.

Following Rudy et al. [12], we will assume $\Theta$ to be overspecified, meaning that the dynamics can be represented as linear combinations of the columns of $\Theta$. However, many PDEs in the natural sciences contain only a few terms. Therefore, we wish to learn a *sparse* vector $\boldsymbol{\xi} = [\xi_1, \ldots, \xi_{N_\ell}]^T$ as a solution of equation (2.7). This is done in PDE-FIND by considering the optimization criterion

$$\bar{\boldsymbol{\xi}} = \text{argmin}_{\boldsymbol{\xi}} \left( ||\Theta(U, Q)\boldsymbol{\xi} - U_t||_2^2 + \lambda ||\boldsymbol{\xi}||_2^2 \right),  \tag{2.11}$$

for the coefficient vector $\boldsymbol{\xi}$, where $\lambda \in \mathbb{R}_{>0}$ is a free parameter that penalizes large coefficients. This is the method of *ridge regression*. We note here that the term $||\boldsymbol{\xi}||_2^2$ can be replaced with $||\boldsymbol{\xi}||_1^2$, which corresponds to performing *LASSO* [8,15]. The optimal choice of implementation is largely problem dependent, and various choices for the regularization method have been compared in the literature [11,12,32,33], although no method has been proven to be definitively preferred over another.

The default implementation of PDE-FIND as proposed by Rudy et al. [12] supplements the ridge regression problem with a *sequential thresholding* procedure in which a solution to equation (2.11) is found, and a hard threshold is performed on the regression coefficients by eliminating all library terms that have coefficients smaller than some pre-specified parameter $d_{\text{tol}}$. This process is then repeated on the remaining library terms until all coefficients are larger than $d_{\text{tol}}$, or until a maximum number of iterations has been reached. The sequential thresholding process is undertaken to enforce sparsity as the solution to the ridge regression problem in equation (2.11) may contain several small, but non-zero values. The combined algorithm is called *sequential thresholding ridge regression* (STRidge). For more details and motivation of the method, we refer to [12], and for completeness we summarize the PDE-FIND method in algorithm 1.

**Algorithm 1** . STRidge.

**Input:** Library matrix, $\Theta(U)$, time derivative of data, $U_t$, STRidge parameters, $\lambda$ and $d_{\text{tol}}$, and maximum number of iterations, *iters*.

**Output:** Sparse vector $\boldsymbol{\xi}$.

1 Set $B = \{j : j = 1, \ldots, N_\ell\}$;

2 **while** *iters* $\geq 0$ **do**

3 Set $\Theta[:, B]$ to be the matrix consisting of all columns of $\Theta(U)$ for which the coefficient $c_j$ has index $j \in B$;

4 Solve the sparse regression problem including only coefficients in $B$, that is, compute $\hat{\boldsymbol{\xi}}[B] = \operatorname{argmin}_{\bar{\boldsymbol{\xi}}} \|\Theta[:, B]\bar{\boldsymbol{\xi}} - U_t\|_2^2 + \lambda\|\bar{\boldsymbol{\xi}}\|_2^2$;

5 Update $B$: set

6 $B = \{j : \text{entry of } \hat{\boldsymbol{\xi}}[B] \text{ corresponding to coefficient } c_j \text{ has magnitude at least } d_{\text{tol}}\}$;

7 $iters \leftarrow iters - 1$;

8 **end**

9 Update $\xi$:
 for $j \notin B$ set the $j$th entry of $\boldsymbol{\xi}$ to be zero;
 for $j \in B$ set the $j$th entry of $\boldsymbol{\xi}$ to be the entry of $\hat{\boldsymbol{\xi}}[B]$ corresponding to coefficient $c_j$.

**Table 3.** Coefficients of the coarse-grained PDEs describing evolution of the mean population density over time for the three example cases used in this work. The coefficients correspond to the coarse-grained PDEs described in table 2.

| case | $c_1$ | $c_u$ | $c_{u^2}$ | $c_{u_x}$ | $c_{u \cdot u_x}$ | $c_{u^2 \cdot u_{xx}}$ | $c_{u_{xx}}$ | $c_{u \cdot u_{xx}}$ | $c_{u^2 \cdot u_{xx}}$ |
|------|-------|-------|-----------|-----------|-------------------|------------------------|--------------|----------------------|------------------------|
| I | 0.0 | 0.0 | 0.0 | 0.0 | 0.0 | 0.0 | 0.25 | 0.0 | 0.0 |
| II | 0.0 | 0.0 | 0.0 | −0.0375 | 0.075 | 0.0 | 0.25 | 0.0 | 0.0 |
| III | 0.0 | 0.01 | −0.01 | 0.0 | 0.0 | 0.0 | 0.25 | 0.0 | 0.0 |

## (i) Application of PDE-FIND to the ABM data

We first generate ABM data for cases I, II and III. For each, we generate two datasets with different noise levels, denoting them $\mathcal{D}_r^i$ where $r = \text{I, II, III}$ denotes the case and $i = 1, 2$ denotes the dataset/noise level. For dataset 1 ($i = 1$), the density profiles are generated using single realizations of the ABM and averaging (so that $K = 1$ and the data are relatively noisy), whereas for dataset 2 ($i = 2$) the density profiles are generated using 50 realizations of the ABM and averaging (so that $K = 50$ and the data contain relatively little noise). For each realization of the ABM, we simulate for $T = 1000$ time steps and subsample the data at every other time point so that each dataset contains information for $n = 500$ time points and $m = 200$ space points. Each dataset contains $N_s$ samples, each an average over $K$ realizations of the ABM. We use the standard implementation of PDE-FIND [12] and use polynomial differentiation at fourth order to evaluate both the time and space derivatives.

 We select a library of candidate terms that includes all polynomial terms up to order 2 and up to the second derivative. Table 3 shows the values of the coarse-grained PDE coefficients, according to equation (2.4). These are the values of the coefficients that we would expect the PDE-FIND algorithm to return for perfect spatio-temporal data. In table 3, and the rest of this work, we use the notation $c_i$ for the coefficient of term $i$ in the learned PDE.

## 3. Sources of variability and model misspecification

In this section, we showcase three different, but related, directions in the uncertainty quantification of the learned differential equations. In §3a, we demonstrate the intrinsic variability

of the learned coefficients in the presence of observation noise. We evaluate how uncertainty changes as the noise level is varied and suggest that even when using state-of-the-art denoising approaches a need for uncertainty quantification remains. Although increasing the signal-to-noise ratio helps, regardless of the method there is still variability that needs to be quantified. In particular, this is important in biological applications where observations are often very noisy and practitioners rarely have access to very large amounts of data. In §3b, we investigate the impact of varying the regularization hyperparameter in PDE-FIND with a view to asking whether this can be optimized to reduce uncertainty. We find that, while this is possible, parameter estimates are still uncertain and this uncertainty needs to be quantified. Finally, in §3c, we demonstrate that a key issue with PDE-FIND is that it aims to fit the time derivative of the solution and does not take into account the fit of the observed density to the data, leading to unphysical predictions. We demonstrate how to mitigate these issues in §4 through the use of Bayesian methods where we can evaluate uncertainty in a framework that optimizes the fit of the model density profile to the data.

In order to quantify the variability in results from the application of PDE-FIND, we introduce a statistic which we term the *identification ratio*. Assume that for each sample, $s$, in the observed dataset (which contains $N_s$ averaged density profile samples) we have used PDE-FIND to produce an estimate of the library coefficients, $\boldsymbol{\xi}$, using STRidge, and we denote this estimate $\hat{\boldsymbol{\xi}}^s$. For each term $i$ in the library, we define the identification ratio, $a_i$, as

$$a_i = \frac{1}{N_s} \sum_{s=1}^{N_s} \mathbb{I}(\hat{\xi}_i^s \neq 0), \tag{3.1}$$

where $\mathbb{I}$ represents the indicator function and $\hat{\xi}_i^s$ is the $i$th entry of $\hat{\boldsymbol{\xi}}^s$. Therefore, $a_i$ quantifies how often the term $\mathcal{N}_i$ from equation (2.7) is included in the PDE-FIND predictions. When $a_i$ is close to unity, the term is identified across many samples as being relevant for the dynamics and, conversely, when $a_i$ is close to zero, the term is identified in only a small minority of samples as being relevant.

## (a) Variability of relevant PDE-FIND coefficients with noisy observations

We first demonstrate that a naive application of PDE-FIND on noisy synthetic data yields variable and unreliable parameter estimates. For this application, we do not carry out hyperparameter tuning, but simply use widely adopted parameter settings to learn the coefficients. For each of the two datasets associated with each of case I, case II and case III, where dataset 1 averages over $K = 1$ realizations and dataset 2 averages over $K = 50$ simulations, we train the PDE-FIND algorithm using algorithm 1 (STRidge) with fixed hyperparameter settings[1] $\lambda = 10^{-2}$ and $d_{\text{tol}} = 10^{-3}$. For each of the resulting datasets, we also compute the corresponding identification ratios (table 4) to quantify the extent of identification of the different terms in the model, and compare the performance of PDE-FIND on the different case studies. In this case, $N_s = 1000$ samples.

### (i) Case I

For case I, recall that the true PDE model contains only the term $u_{xx}$ with coefficient 0.25, hence in noise-free scenarios we anticipate that $c_{u_{xx}}$ should be non-zero and all other coefficients should be zero. Table 4 shows that the two terms identified regularly by PDE-FIND on $\mathcal{D}_I^1$, the high-noise dataset, are $u_{xx}$ and $uu_{xx}$, with identification ratios of 0.826 and 0.199, respectively. On $\mathcal{D}_I^2$, the low-noise dataset, $u_{xx}$ is consistently identified and no other terms are identified. However, there is significant variability in the learned coefficients between different samples from the same dataset (figure 3). In addition, for the high-noise dataset, $\mathcal{D}_I^1$, the coefficients of $u_{xx}$ and $uu_{xx}$ are correlated (figure 3c). In some cases, PDE-FIND identifies just one of the two terms, and in others

---

[1]These settings were used in the context of estimating the diffusion parameter in a random walk model in the electronic supplementary material of [12].

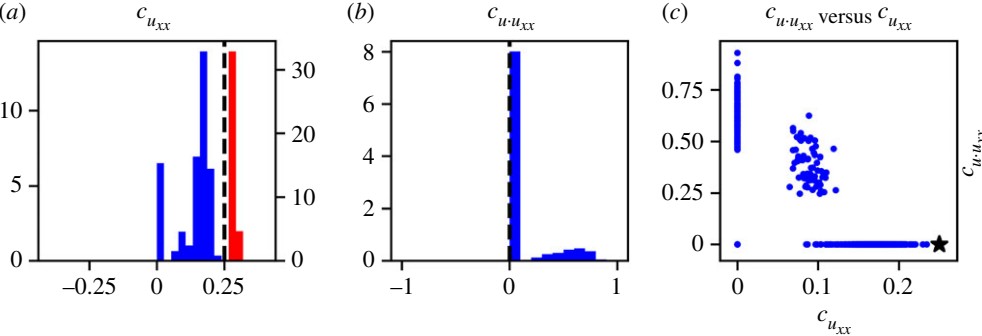

**Figure 3.** Histograms showing the empirical distribution of relevant PDE-FIND coefficients for case I. (*a*) Histograms for $c_{u_{xx}}$ generated using $\mathcal{D}_I^1$ (blue) and $\mathcal{D}_I^2$ (red) compared with the true parameter value (black line). (*b*) Histogram for $c_{u \cdot u_{xx}}$ generated using $\mathcal{D}_I^1$ compared with the true value (black line). (*c*) Joint distribution of $c_{u_{xx}}$ and $c_{u \cdot u_{xx}}$ generated using $\mathcal{D}_I^1$ compared with the true parameter (black star). (Online version in colour.)

**Table 4.** Identification ratios for different datasets. Italic fonts indicate terms that we anticipate in the learned PDEs based on the results from ABM coarse-graining. Recall that case I includes non-biased motility and no proliferation, case II includes motility bias but no proliferation and case III includes non-biased motility and proliferation. Dataset 1 contains averages over $K = 1$ realizations while dataset 2 contains averages over $K = 50$ simulations.

| experiment | $c_1$ | $c_u$ | $c_{u^2}$ | $c_{u_x}$ | $c_{u \cdot u_x}$ | $c_{u^2 \cdot u_x}$ | $c_{u_{xx}}$ | $c_{u \cdot u_{xx}}$ | $c_{u^2 \cdot u_{xx}}$ |
|---|---|---|---|---|---|---|---|---|---|
| $\mathcal{D}_I^1$ | 0.001 | 0.0 | 0.002 | 0.0 | 0.008 | 0.008 | *0.826* | 0.199 | 0.05 |
| $\mathcal{D}_I^2$ | 0.0 | 0.0 | 0.0 | 0.0 | 0.0 | 0.0 | *1.0* | 0.0 | 0.0 |
| $\mathcal{D}_{II}^1$ | 0.0 | 0.0 | 0.0 | *0.999* | *0.0* | 0.0 | *0.012* | 0.002 | 0.0 |
| $\mathcal{D}_{II}^2$ | 0.0 | 0.0 | 0.0 | *1.0* | *0.0* | 0.0 | *0.0* | 0.0 | 0.0 |
| $\mathcal{D}_{III}^1$ | 0.013 | *0.659* | 0.482 | 0.002 | 0.002 | 0.007 | *0.571* | 0.01 | 0.014 |
| $\mathcal{D}_{III}^2$ | 0.0 | *1.0* | 0.0 | 0.0 | 0.0 | 0.0 | *1.0* | 0.0 | 0.0 |
| subsampling $\mathcal{D}_I^1$ | 0.0 | 0.0 | 0.003 | 0.001 | 0.002 | 0.007 | *0.998* | 0.365 | 0.063 |

it identifies a combination of the two. This result highlights that potentially the wrong PDE can be learnt from noisy data, partly because of the fact that different PDEs can give rise to similar predictions. Since all of the ABM data samples have the same corresponding coarse-grained PDE, this highlights the inability of PDE-FIND to confidently learn the governing PDE from noisy data.

### (ii) Case II

For case II, where motility is biased, table 4 shows that the two terms identified regularly by PDE-FIND on the high-noise dataset $\mathcal{D}_{II}^1$ are $u_x$ and $u_{xx}$, with identification ratios of 0.999 and 0.012, respectively. Note that the true model should also contain the term $uu_x$, but PDE-FIND fails to identify this term across all the samples of the dataset. This term arises in the coarse-grained PDE as a result of volume exclusion (incorporated into the ABM through the requirement that at most one agent can occupy a lattice site at any instant in time). Therefore, we infer in this case that the data are insufficient to identify the impact of volume exclusion. This is most likely a result of the initial conditions and/or the time scale over which data are collected since the density is relatively low across the domain and so crowding is likely to be unimportant.

For the low-noise dataset $\mathcal{D}_{II}^2$ only $u_x$ is identified, with an identification ratio of 1.0. The histograms in figure 4 reveal a significant amount of variability in the learned parameters. For instance, for both $\mathcal{D}_{II}^1$ and $\mathcal{D}_{II}^2$, the parameters are distributed far away from the true parameter

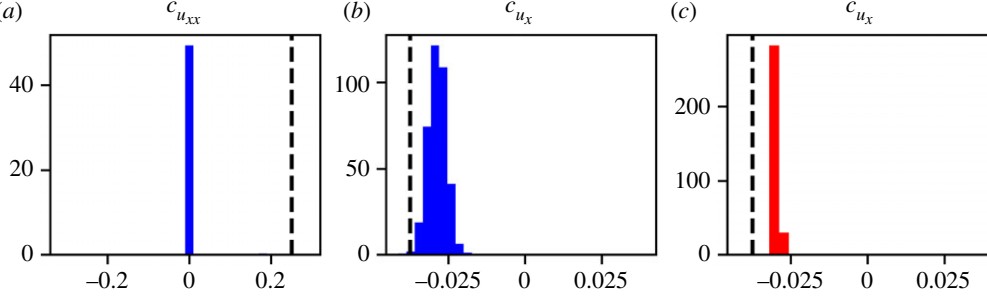

**Figure 4.** Histograms showing the empirical distribution of relevant PDE-FIND coefficients for case II, in which motility is biased but there is no proliferation. (*a*) Histogram for $c_{u_{xx}}$ generated using $\mathcal{D}_{\mathrm{II}}^1$ (blue) compared with the true parameter value (black line). (*b*) Histogram of $c_{u_x}$ generated using $\mathcal{D}_{\mathrm{II}}^1$ (blue) compared with the true parameter value (black line). (*c*) Histogram of $c_{u_x}$ generated using $\mathcal{D}_{\mathrm{II}}^2$ (red) compared with the true parameter value (black line). (Online version in colour.)

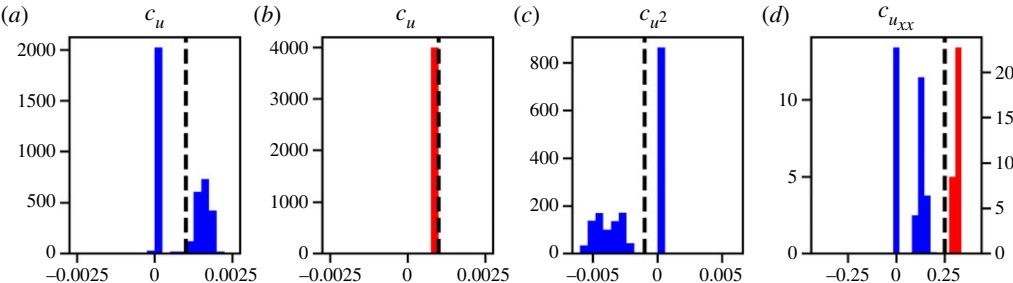

**Figure 5.** Histograms showing the empirical distribution of relevant PDE-FIND coefficients for case III, which includes random motility and proliferation. (*a*) Histogram for $c_u$ trained on $\mathcal{D}_{\mathrm{III}}^1$ (blue) compared with the true parameter value (black line). (*b*) Histogram for $c_u$ trained on $\mathcal{D}_{\mathrm{III}}^2$ (red) compared with the true parameter value (black line). (*c*) Histogram for $c_{u^2}$ trained on $\mathcal{D}_{\mathrm{III}}^1$ (blue) compared with the true parameter value (black line). (*d*) Histogram for $c_{u_{xx}}$ trained on $\mathcal{D}_{\mathrm{III}}^1$ (blue) and $\mathcal{D}_{\mathrm{III}}^2$ (red) compared with the true parameter value (black line). (Online version in colour.)

value. The variability appears to decrease with the noise level, and the distribution of estimated parameters moves towards the true parameter value; however, the $u_{xx}$ coefficient is 'lost' in the process.

### (iii) Case III

For case III, which includes proliferation, table 4 shows that the terms identified regularly by PDE-FIND on the high-noise dataset $\mathcal{D}_{\mathrm{III}}^1$ are $u$, $u^2$ and $u_{xx}$ with identification ratios equal to 0.659, 0.482 and 0.571, respectively. Note that this means that all terms we would expect to appear in the PDE are identified. However, as shown in detail in the electronic supplementary material, figure S6, there is a correlation between the learned coefficients of $u$ and $u^2$, which points towards non-identifiability [34,35]. For the low-noise dataset $\mathcal{D}_{\mathrm{III}}^2$, the parameters identified are $u$ and $u_{xx}$, both with identification ratio equal to 1.0. The histograms in figure 5 reveal that the variability in the learned coefficients decreases as the noise level in the data is decreased. However, this does not mean that the model is increasingly well identified as the noise is decreased—the term $u^2$ is not identified by PDE-FIND for the low-noise dataset $\mathcal{D}_{\mathrm{III}}^2$, which contradicts the mean-field analysis.

### (iv) Methods to decrease the noise levels

To investigate the impact of noise on PDE-FIND, we carried out two further studies in which the noise in the data is reduced. First, we investigated whether choosing a more coarse spatial grid improves the PDE-FIND predictions. Choosing a more coarse spatial discretization results

in a smoother density profile; however, greater errors are incurred in the approximation of the spatial derivatives and fewer data points are available. For this experiment, we subsampled the data along the $x$-dimension by averaging the occupancy over multiple columns at a time. Mathematically, from the empirical densities, $C_i$, at each time point, we subsample over intervals containing $2B$ lattice sites, estimating the average occupancies $\tilde{C}_i$ for $1 \le i \le I/(2B)$ as

$$\tilde{C}_i = \frac{1}{2B} \sum_{\ell=2B(i-1)+1}^{2Bi} C_\ell. \tag{3.2}$$

Table 4 summarizes the identification ratios found for the high-noise dataset $\mathcal{D}_I^1$, where motility is unbiased and there is no proliferation, and we take $B = 2$. We see that with spatial subsampling the identification ratio of $c_{u_{xx}}$ increases significantly, although there is no marked improvement in the identification ratios of other terms in the model. We conclude that, even if this method of noise reduction allows the correct coefficients to be identified more frequently, there remains a need to mitigate the fact that many other terms are spuriously identified by PDE-FIND.

Second, we investigated other means to reduce observation noise. In applications of PDE-FIND to real-life data, practitioners will typically not be able to control for the amount of observation noise in the way that is done in the numerical experiments of this work, hence methods to smooth data may be useful in allowing identification of the PDE model. In electronic supplementary material, §S3, we explore two practically appealing methods, convolution with a Gaussian kernel and an implementation of principal component analysis for EQL by Li *et al.* [22]. Our results show that, even with these well-established techniques for reducing the influence of noise, predictions remain variable with coefficients highly correlated, and that uncertainty quantification remains necessary for a reliable application of PDE-FIND to realistic biological data.

## (b) Role of the regularization hyperparameter

Recall that, in algorithm 1, a free parameter, $\lambda$, controls the level of penalty incurred by choosing large coefficients in the solution of equation (2.7). It is well known that the choice of regularization parameter is non-trivial because it modulates the amount of sparsity that is enforced on the estimated coefficients. The issue of how to choose the optimal value of this hyperparameter in the context of ABMs was addressed recently by Nardini *et al.* [8], who discussed cross-validation, among other options. As a test case to investigate the effects of algorithm hyperparameters on the uncertainty of learned coefficients, we perform cross-validation on the dataset $\mathcal{D}_I^1$ and then apply PDE-FIND using the optimal value of $\lambda$ found. To do this, we apply the grid search implementation of cross-validation suggested in [8], as detailed in electronic supplementary material, §S4, to arrive at an optimal value of $\lambda = 0.5$. We note here that this value is problem dependent, and whenever a new dataset is being investigated a different value of $\lambda$ will generally be appropriate.

While the results presented in electronic supplementary material, §S4 show that cross-validation improves the performance of PDE-FIND dramatically, as the number of misspecified coefficients decreases sharply when the regularization parameter is optimized, cross-validation does not provide a sufficient solution to manage the uncertainty associated with variability in the predicted coefficients. Figure 6 shows that, even with the optimal value of the regularization coefficient, there is still much uncertainty in the coefficients, as the support of the histogram is large. While the atom at zero has nearly vanished, uncertainty quantification is still necessary because the empirical distribution still indicates a large degree of variability. Moreover, figure 6 shows that at the optimal value of the tuning parameter, $\lambda$, the coefficients $c_{u_{xx}}$ and $c_{u \cdot u_{xx}}$ still have a non-trivial joint distribution, implying that, even with an optimal choice of the regularization parameter, Bayesian methods are needed to analyse the joint behaviour of these two coefficients.

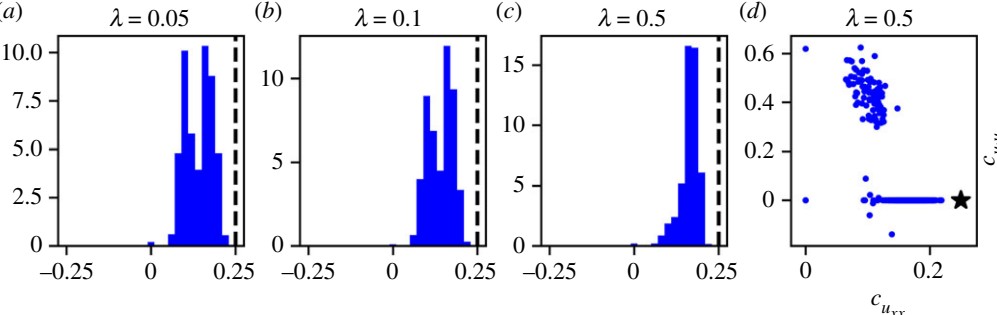

**Figure 6.** Empirical distributions of relevant PDE-FIND coefficients learnt using $\mathcal{D}_I^1$, which consists of unbiased motility only. (a) Histogram of $c_{u_{xx}}$ coefficients generated using $\lambda = 0.01$, compared with the true value 0.25 (black line). (b) Histogram of $c_{u_{xx}}$ coefficients generated using $\lambda = 0.05$, compared with the true value 0.25 (black line). (c) Histogram of $c_{u_{xx}}$ coefficients generated using the optimal value of $\lambda = 0.5$, compared with the true value 0.25 (black line). (d) Empirical joint distribution of $c_{u_{xx}}$ and $c_{u \cdot u_{xx}}$ coefficients generated using the optimal value of $\lambda = 0.5$, compared with the true parameters $(c_{u_{xx}}, c_{u \cdot u_{xx}}) = (0.25, 0.0)$ (black star). (Online version in colour.)

## (c) Comparison of model predictions

We now provide an explanation for the poor performance of PDE-FIND on the ABM data. The PDE-FIND algorithm solves a sparse regression problem to fit linear combinations of spatial derivatives to the time derivative. When data contain little to no noise, the temporal and spatial derivatives can be accurately estimated, and so the relationship between spatial and temporal derivatives can be inferred from observed data. In this context, comparing model predictions by their performance with respect to the $L^2$-loss in the learned temporal derivative retrieves the ground truth.[2] However, when the data are noisy, a number of different linear combinations of the spatial derivatives can result in an $L^2$-loss comparable to (or better than) those of the ground truth PDE (figure 7). In electronic supplementary material, §S5 (figures S11–S13), we demonstrate this by selecting, for each dataset, two instances where the learned equations contain different terms from the coarse-grained PDE, yet in both cases the temporal derivatives reproduce the observed temporal derivative qualitatively. However, there is no guarantee that such a match in the temporal derivative is sufficient to yield solutions that resemble the observed data when the PDE is numerically evaluated. We illustrate this in figure 8, where, for each of cases I, II and III, we select two sets of coefficients learned by PDE-FIND: one where the solution of the corresponding PDE resembles a typical data trace, and one where the solution of the corresponding PDE bears little resemblance to typical observed data traces.

In summary, what this striking difference in predictive capabilities of the learned PDEs reveals is that coefficients that optimize equation (2.11) do not necessarily perform well in terms of their ability to predict the evolution of the spatio-temporal density profiles. We illustrate this in further detail in figure 7. We take the dataset $\mathcal{D}_I^1$, which consists of unbiased motility and no proliferation; each sample in the dataset consists of an average over $K = 1$ simulations from the ABM. First, we average over all $N_s = 1000$ samples in the dataset to obtain the density profile $\langle C_i(t) \rangle$ as in equation (2.2). The two coefficients consistently identified for dataset $\mathcal{D}_I^1$ are $c_{u_{xx}}$ and $c_{u \cdot u_{xx}}$, which give the PDE

$$u_t = c_{u \cdot u_{xx}} u u_{xx} + c_{u_{xx}} u_{xx}. \tag{3.3}$$

We integrate this PDE numerically over a grid of values of $c_{u_{xx}}$ and $c_{u \cdot u_{xx}}$, and then evaluate the $L^2$-loss between the time derivative of the PDE model and that of the averaged ABM data, $\langle C_i(t) \rangle$ (figure 7a), as well as the difference between the density predicted by the PDE model and that of the averaged ABM data (figure 7b). We estimate the sum of the $L^2$-loss between the PDE and

---

[2]For functions $f$ and $g$, the $L^2$-loss is given by $||f - g||_2^2 = \int [f(x) - g(x)]^2 \, dx$.

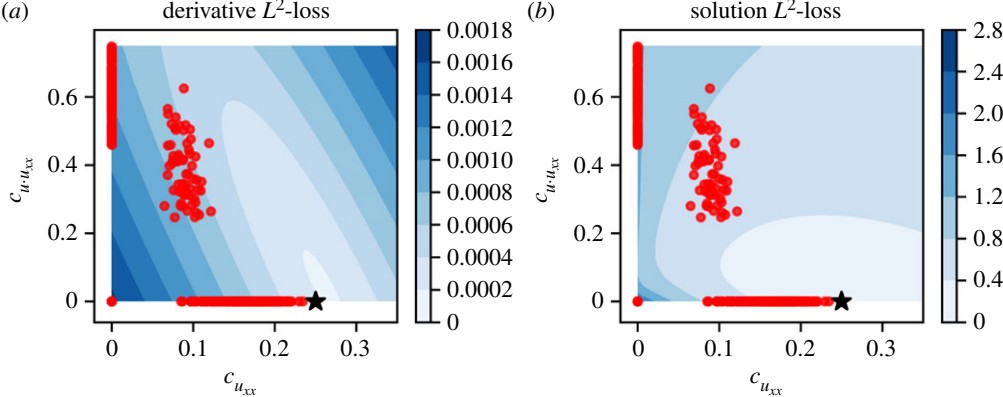

**Figure 7.** Heatmaps showing differences between averaged ABM data and the PDE solution in equation (3.3). (*a*) $L^2$-loss landscape of the time derivative, estimated using equation (3.4). (*b*) $L^2$-loss landscape for the density profile, estimated using equation (3.4). In each plot, the pairs of coefficients estimated by applying PDE-FIND to each of the $N_s = 1000$ samples of the dataset are plotted using red dots, and the parameter values used to generated the ABM data are indicated using a black star. (Online version in colour.)

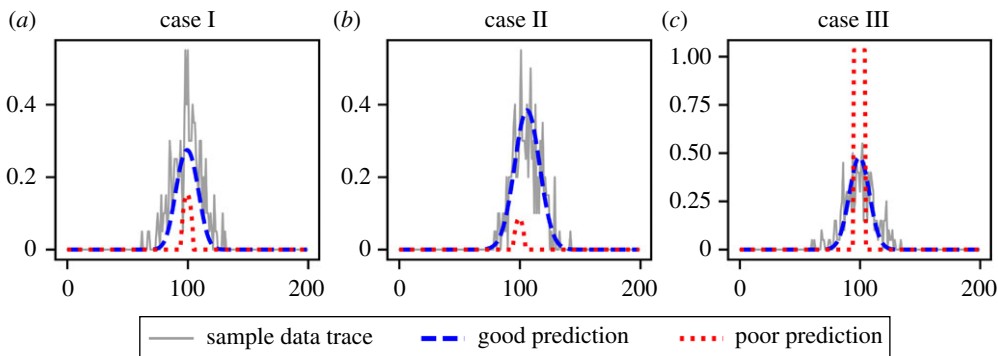

**Figure 8.** Comparison of predictions made by misspecified PDEs that are learned through the application of PDE-FIND to noisy data. (*a*) Case I: using $c_{u_{xx}} = 0.104$, $c_{u \cdot u_{xx}} = 0.26$ (blue dash) and $c_{u^2 \cdot u_{xx}} = 1.517$ (red dots). (*b*) Case II: using $c_{u_x} = -0.0264$, $c_{u_{xx}} = 0.20$ (blue dash) and $c_{u \cdot u_{xx}} = 0.641$ (red dots). (*c*) Case III: using $c_{u_{xx}} = 0.12$ (blue dash) and $c_u = 0.0130$ (red dots). All other coefficients are set to zero. (Online version in colour.)

ABM data at five time points as

$$d(X^{\text{obs}}, X^{\text{sim}}) = \sum_{j=1}^{5} ||X^{\text{obs}}_{50j} - X^{\text{sim}}_{50j}||_2, \tag{3.4}$$

where, for example when comparing density profiles,

$$X^{\text{obs}}_{50j} = \left[ \langle C_1(50j) \rangle, \langle C_2(50j) \rangle, \ldots, \langle C_{200}(50j) \rangle \right]^T, \tag{3.5}$$

$$X^{\text{sim}}_{50j} = [u(\Delta, 50j), u(2\Delta, 50j), \ldots, u(200\Delta, 50j)]^T, \tag{3.6}$$

and when comparing time derivatives,

$$X^{\text{obs}}_{50j} = \left[ \langle C_{1t}(50j) \rangle, \langle C_{2t}(50j) \rangle, \ldots, \langle C_{200t}(50j) \rangle \right]^T, \tag{3.7}$$

$$X^{\text{sim}}_{50j} = [u_t(\Delta, 50j), u_t(2\Delta, 50j), \ldots, u_t(200\Delta, 50j)]^T, \tag{3.8}$$

with $\langle C_{it}(50j)\rangle = (\langle C_i(50j)\rangle - \langle C_i(50j+\tau)\rangle)/\tau$ for $j=1,\ldots,5$, where $\tau$ is the time step of the ABM simulation algorithm. The blue shading in figure 7 shows the $L^2$-loss in each case, and we also plot the PDE-FIND-estimated coefficients on the same axes for each of the $N_s = 1000$ samples of the dataset.

Figure 7 demonstrates that there are significant differences between the loss landscapes of the different error metrics. While both loss landscapes show a minimum around the parameter set used in the ABM simulations ($c_{u_{xx}} = 0.25$ and $c_{u \cdot u_{xx}} = 0.0$; black stars in figure 7), the derivative loss landscape in figure 7 is unable to distinguish different regions of $(c_{u_{xx}}, c_{u \cdot u_{xx}})$ parameter space. For example, there are multiple PDE-FIND parameter sets (red dots) that lie close to the $L^2$-loss contours of 0.0010 and 0.0012; some of these sit on the horizontal axis (where $c_{u_{xx}}$ is non-zero and $c_{u \cdot u_{xx}}$ is zero), whereas others sit on the vertical axis of the plot (where $c_{u_{xx}}$ is zero and $c_{u \cdot u_{xx}}$ is non-zero). On the other hand, the $L^2$-loss for the density profiles provides much more useful information: as one moves further from the input parameter values ($c_{u_{xx}} = 0.25$ and $c_{u \cdot u_{xx}} = 0.0$; black star in figure 7) there are increasing errors between the density profile predicted by solution of the PDEs and the averaged ABM data. Electronic supplementary material, figure S14 demonstrates that this issue is further compounded as the noise in the data increases.

In summary, in this section we have shown that the density profile loss landscape is much more informative about the underlying PDE model than the derivative loss landscape. We exploit this observation in the following section, where we propose a method which we term 'Bayes-PDE-FIND' both to tackle the issues relating to the uncertainty in the predictions of PDE-FIND for noisy data and to quantify the uncertainty in PDE-FIND predictions.

# 4. Bayes-PDE-FIND

In this section, we propose an approach that harnesses the likelihood-free method of approximate Bayesian computation to quantify the uncertainty in estimates provided by PDE-FIND. In brief, our method involves the application of PDE-FIND to multiple datasets in order to define a prior distribution for the coefficients of the PDE library terms, $\mathcal{N}_i$ for $i = 1, \ldots, N_\ell$, followed by application of Bayesian approaches for estimation of the posterior parameter distribution.

## (a) Approximate Bayesian computation

The goal of Bayesian parameter estimation is to update prior beliefs about model parameters $\boldsymbol{\theta}$ encoded in a prior distribution $\pi(\boldsymbol{\theta})$. In this context, $\boldsymbol{\theta}$ constitutes the coefficients, $\xi_i$, of the library terms $\mathcal{N}_i$ for $i = 1, \ldots, N_\ell$. The updating process is dependent on observations $\mathcal{D}_{\text{obs}}$, which in this work are the noisy, averaged data from the ABM, as detailed in §2i. The mathematical model is the PDE defined in equation (2.7), which then defines a likelihood $P(\mathcal{D}_{\text{obs}} \mid \boldsymbol{\theta})$. In Bayesian statistics, the likelihood is combined with the prior distribution to give the posterior distribution,

$$P(\boldsymbol{\theta} \mid \mathcal{D}_{\text{obs}}) \propto P(\mathcal{D}_{\text{obs}} \mid \boldsymbol{\theta})\pi(\boldsymbol{\theta}). \tag{4.1}$$

Such posterior distributions provide information as to the uncertainty in parameter estimates that are learned from observed data, and also allow practitioners to understand the range of realistic parameter values that can produce observed data. The likelihood $P(\mathcal{D}_{\text{obs}} \mid \boldsymbol{\theta})$ defines the probability density of the observations $\mathcal{D}_{\text{obs}}$ given the model parameters $\boldsymbol{\theta}$. In this context, the observed data $\{U(x,t)\}$ at space points $x = x_1, x_2, \ldots, x_N$ and time points $t = t_1, t_2, \ldots, t_N$ are obtained from a stochastic ABM. The solution $u(x,t; \boldsymbol{\theta})$ of the PDE model is an approximation of the mean of the ABM data, i.e. $u(x,t; \boldsymbol{\theta}) \approx \mathbb{E}_{\boldsymbol{\theta}}[U(x,t)]$. To define a classical likelihood, one would need to first assume that the mean of the ABM data is exactly given by the PDE solution and prescribes the distribution of ABM outputs around the PDE model mean. However, for a general ABM, the distribution of the deviation from the mean is unknown. In some cases, one might choose to make a simplifying assumption, such as a Gaussian approximation. However, in the small data limit considered in EQL applications, such an assumption is unreasonable. It will depend on the details of the ABM as to the extent to which individual realizations vary from

their mean, which is, for the purposes of inference, unknown. As we prefer to avoid placing unnecessary assumptions on the process, we opt instead for a likelihood-free approach. We provide further mathematical insight and justification for avoiding likelihood-based methods in electronic supplementary material, §S7.

Approximate Bayesian computation is a popular likelihood-free tool to estimate the posterior parameter distribution [36,37]. It approximates the likelihood, $P(\mathcal{D}_{\mathrm{obs}} \mid \boldsymbol{\theta})$, using repeated simulation of the model, and acceptance of the parameter $\boldsymbol{\theta}$ requires that the output of the model, $\mathcal{D}_{\mathrm{sim}}(\boldsymbol{\theta})$, is in some sense close enough to the data, $\mathcal{D}_{\mathrm{obs}}$. The ABC posterior can be written

$$P_{\mathrm{ABC}}(\boldsymbol{\theta} \mid \mathcal{D}_{\mathrm{obs}}) \propto P(d(\mathcal{D}_{\mathrm{obs}}, \mathcal{D}_{\mathrm{sim}}) < \varepsilon \mid \boldsymbol{\theta})\pi(\boldsymbol{\theta}), \qquad (4.2)$$

where $d$ is a distance function that quantifies the difference between the data, $\mathcal{D}_{\mathrm{obs}}$, and model output, $\mathcal{D}_{\mathrm{sim}}$. In this work, our aim is to use ABC to estimate the posterior distribution of the PDE model coefficients, $\xi_i$ for $i = 1, \dots, N_\ell$, for a given dataset.

Despite the apparent simplicity of the ABC method, unless an informative prior is used to constrain the space of possible parameters and informative summary statistics can be found, the application of ABC methods to models with high-dimensional parameter space and output space is generally computationally prohibitive. In particular, this high computational cost means that the direct application of ABC methods to estimate the coefficients $\xi_i$, for $i = 1, \dots, N_\ell$, in equation (2.7) is essentially infeasible unless it is possible to construct an informed prior distribution. Here, we propose a method that uses the predictions of PDE-FIND to construct an informed prior so that ABC can then be used to estimate the library coefficients $\xi_i$.

## (b) Using PDE-FIND to define a prior distribution for ABC

Assume that, for each sample, $s$, in the observed dataset (which contains $N_s$ averaged density profile samples), we have used PDE-FIND (as defined using algorithm 1) to produce an estimate, $\hat{\boldsymbol{\xi}}^s$, of the parameters $\boldsymbol{\xi}$. Recall that, for each term $i$ in the library, we have defined the identification ratio, $a_i$, as

$$a_i = \frac{1}{N_s} \sum_{s=1}^{N_s} \mathbb{I}(\hat{\boldsymbol{\xi}}_i^s \neq 0)$$

to quantify how often the term $\mathcal{N}_i$ from equation (2.7) is included in the PDE-FIND predictions.

To make progress in specifying a prior distribution for the library coefficients, $\xi_i$, we first threshold, using parameter $0 < \delta < 1$, so that we can define $A = \{i : a_i > \delta\}$ as the set of coefficients that are identified by PDE-FIND in more than a fraction $\delta$ of the $N_s$ samples of the dataset. We then eliminate from the library all terms for which $a_i < \delta$, i.e. we set the marginal prior for coefficient $\xi_i$ to be $\pi_i \equiv 0$. This achieves a first step of coefficient selection by eliminating variables for which the initial PDE-FIND screen indicates low confidence. On the other hand, for $i \in A$, there is still a need to investigate which coefficients to include in the final model, and a prior must be carefully chosen to explore which parameters to include and which parameters to eliminate.

Spike-and-slab models are powerful tools to perform variable selection in regression problems [38–41]. The main idea of a spike-and-slab-type prior is that it defines a two-point mixture distribution in which coefficients are mutually independent. Each mixture is made up of a flat distribution with large support (the slab) and a degenerate distribution at zero (the spike). In early formulations, the slab was modelled as a uniform distribution over some region of parameter space [39,40], whereas, in more recent work, inference is performed on hyperparameters of the marginal distributions [38,41]. Samples of the hyperparameters yielding a high variance will lead to sampling parameters far away from zero, whereas samples of the hyperparameters yielding a low variance will sample close to zero. In this way, the aim is to explore parameter space by iteratively sampling over the hyperparameters and the values for the coefficients using Gibbs sampling. In this work, we wish to exploit the simplicity of the earliest slab-and-spike models, which use a Dirac measure at zero to enforce sparsity, while using as much information as possible

from the PDE-FIND screen in defining the prior distribution without violating the likelihood principle.

We follow the hierarchical Bayesian group LASSO model with an independent spike-and-slab-type prior for each coefficient [41]. We set the group size in the model of Xu & Ghosh [41] equal to 1, so that the prior $\pi_i(\xi_i)$ for each coefficient $\xi_i$ is given by

$$\xi_i | \mu, \sigma_i^2 \sim (1 - a_i)\delta_0 + a_i \mathcal{N}(\mu_i, \sigma_i^2),$$

$$\sigma_i^2 \sim \mathcal{IG}(\alpha_i, \beta_i),$$

where $\mathcal{IG}$ is the inverse-gamma distribution with parameters $\alpha_i, \beta_i$ that define the shape of the prior on $\sigma_i^2$. This is the standard choice for modelling the distribution of the hypervariances. In the approach of Xu & Ghosh [41], $\mu = 0$. In this work, we can use knowledge of the coefficients gained through our initial PDE-FIND screen to inform the $\mu_i$. To do so, we randomly divide the ABM data in half and use one subset in the PDE-FIND screen to inform the prior (*exploration subset*) and the other subset to perform inference (*inference subset*). We first set $a_i$ equal to the $i$th identification ratio and $\mu_i$ equal to the $i$th sample mean of the PDE-FIND coefficients trained on the exploration subset. Since the hyperprior for the variances $\sigma_i^2$ allows for large values of $\sigma_i^2$, this prior is not overly restrictive, since values far away from the sample mean can be sampled. Second, we tune $\alpha_i, \beta_i$ using the exploration subset so that the variance is on average the same order of magnitude as the PDE-FIND coefficients. This is crucial: a large (small) variance in parameters that are typically small (large) will fail to sample from the relevant regions of parameter space. The exact values of $\mu_i, \alpha_i, \beta_i$ are given in electronic supplementary material, §S9.

These considerations now imply that the prior for $\boldsymbol{\xi}$ is given by

$$\boldsymbol{\pi} = \bigotimes_{i=1}^{N_\ell} \left\{ \mathbb{I}(i \in A)\pi_i \xi_i + \mathbb{I}(i \notin A) \cdot \delta_0 \right\}. \tag{4.3}$$

The key advantage in specifying this prior distribution is that we are now only required to perform Bayesian inference for a reduced model that has a much lower dimensional parameter space (equal to $|A| \ll N_\ell$) than the original model (with parameter space of dimension $N_\ell$). This is because PDE-FIND promotes sparsity and so the majority of coefficients will have identification ratio, $a_i$, close to zero. Such terms can be confidently eliminated from the model and so not considered in the ensuing parameter estimation process. On the other hand, if $a_i \approx 1$ this means PDE-FIND has consistently identified the $i$th term as relevant for the dynamics. As such, we can be confident that $\mathcal{N}_i$ should be included in the model; however, uncertainty in estimates of $\xi_i$ must still be quantified. Finally, if $a_i$ is close to neither zero nor 1, which means that PDE-FIND has included the $i$th term in the library for a non-trivial number of samples in the dataset, Bayesian approaches can be used to investigate the joint posterior distribution of the $i$th coefficient, $\xi_i$, with the rest of the model terms by considering the performance of models that both include and exclude the $i$th term.

## (c) The Bayes-PDE-FIND algorithm

We now outline the Bayes-PDE-FIND approach. In essence, we apply the PDE-FIND algorithm to each sample $s = 1, \ldots, N_s$ of the dataset under consideration, and use the results to formulate a prior distribution for ABC as described in §4b and, in particular, in equations (4.2) and (4.3). We then apply ABC to estimate the posterior parameter distribution, noting that the computational cost of ABC is much reduced through the use of PDE-FIND to generate an informed prior distribution—in effect, we use PDE-FIND to reduce the target PDE in equation (2.7) to

$$u_t = \sum_{i \in A} \mathcal{N}_i(u, u_x, u_{xx}, \ldots) \, \xi_i, \tag{4.4}$$

with a prior distribution over the $\xi_i$, for $i \in A$, given by equation (4.3).

Importantly, when we apply ABC to estimate the posterior parameter distribution, we use a low-noise dataset, $\langle C_i(t)\rangle_{\mathrm{LN}}$ for $1 \le i \le I$, created by averaging over the inference subset of the dataset as the observed data, $\mathcal{D}_{\mathrm{obs}}$, where sample $s$ is calculated as

$$\langle C_i(t)\rangle_s = \sum_{k=1}^{K} C_i^{k,s}(t) = \frac{1}{J \cdot K} \sum_{k=1}^{K} \sum_{j=1}^{J} C_{ij}^{k,s}(t), \tag{4.5}$$

that is, each sample $s$ consists of column-averaged data from $K$ simulations of the ABM, and so, for $1 \le i \le I$,

$$\langle C_i(t)\rangle_{\mathrm{LN}} = \sum_{s=1}^{S} \langle C_i(t)\rangle_s. \tag{4.6}$$

For the distance function, $d$, we again use an averaged estimate of the $L^2$-difference between the ABM data and the PDE solution, here defined as

$$d(\mathcal{D}_{\mathrm{obs}}, \mathcal{D}_{\mathrm{sim}}(\boldsymbol{\theta})) = \sum_{j=1}^{5} ||(\mathcal{D}_{\mathrm{obs}})_{50j} - \mathcal{D}_{\mathrm{sim}}(\boldsymbol{\theta})_{50j}||_2, \tag{4.7}$$

where, for $j = 1, \ldots, 5$,

$$(\mathcal{D}_{\mathrm{obs}})_{50j} = \left[ \langle C_1(50j)\rangle_{\mathrm{LN}}, \langle C_2(50j)\rangle_{\mathrm{LN}}, \ldots, \langle C_{200}(50j)\rangle_{\mathrm{LN}} \right]^T, \tag{4.8}$$

$$\mathcal{D}_{\mathrm{sim}}(\boldsymbol{\theta})_{50j} = [u(\Delta, 50j; \boldsymbol{\theta}), u(2\Delta, 50j; \boldsymbol{\theta}), \ldots, u(200\Delta, 50j; \boldsymbol{\theta})]^T, \tag{4.9}$$

and $u(x, t; \boldsymbol{\theta})$ is the solution to equation (4.4) with parameter set $\boldsymbol{\theta}$. Note that we choose in equation (4.7) to compare the solutions at a wide range of time points to capture the behaviours of the data over different time scales. We summarize the Bayes-PDE-FIND algorithm in algorithm 2.

---

**Algorithm 2** . Bayes-PDE-FIND.

---

**Input:** Time-series dataset consisting of $N_s$ samples; PDE-FIND hyperparameters $\lambda$ and $d_{\mathrm{tol}}$; PDE library $\mathcal{N}_i$ for $i = 1, \ldots, N_\ell$; minimum identification ratio $\delta > 0$.
**Output:** Posterior distribution over coefficients $\xi_i$, for $i = 1, \ldots, N_\ell$, of library PDE.
1  **for** $s = 1, \ldots, N_s$ **do**
2      Compute $\hat{\boldsymbol{\xi}}^s$ using algorithm 1 with sample $s$ from the dataset.
3  **end**
4  **for** $i \in 1, \ldots, N_\ell$ **do**
5      Compute the identification ratio

$$a_i = \frac{1}{N_s} \sum_{s=1}^{N_s} \mathbb{I}\left(\hat{\boldsymbol{\xi}}_i^s \ne 0\right).$$

6  **end**
7  Compute $A = \{i : a_i > \delta\}$, and define the prior distribution $\boldsymbol{\pi}$ as in equation (4.3):
8

$$\boldsymbol{\pi} = \bigotimes_{i=1}^{N_\ell} \left\{ \mathbb{I}(i \in A)\pi_i \xi_i + \mathbb{I}(i \notin A) \cdot \delta_0 \right\}. \tag{4.10}$$

9  Perform ABC using observed data $\mathcal{D}_{\mathrm{obs}} = [\langle C_1(t)\rangle_{\mathrm{LN}}, \ldots, \langle C_{200}(t)\rangle_{\mathrm{LN}}]$ to obtain the posterior distribution.

---

## (d) Results

The aim of this section is to showcase how the Bayes-PDE-FIND algorithm can be used to significantly improve the quality of the learned PDE model. Recall that the aim is to reduce the uncertainty surrounding which coefficients to include in the model, to reduce uncertainty in the estimated model parameters and to improve the posterior predictive capability of the model by finding a posterior parameter distribution that takes into account properties of the observed density profiles. For each of the noisy-data test cases, that is, datasets $\mathcal{D}_I^1$, $\mathcal{D}_{II}^1$ and $\mathcal{D}_{III}^1$, we apply algorithm 2, using the Pakman package [42] for the ABC step.

Recall that for case I, where the ABM contains only unbiased motility, the only two coefficients regularly identified using PDE-FIND are $c_{u_{xx}}$ and $c_{u \cdot u_{xx}}$ (table 4). This means that all library terms except for $u_{xx}$ and $uu_{xx}$ can be confidently excluded, and for the ABC process we consider the PDE model

$$u_t = c_{u_{xx}} u_{xx} + c_{u \cdot u_{xx}} uu_{xx}, \tag{4.11}$$

and aim to infer the $(c_{u_{xx}}, c_{u \cdot u_{xx}})$ posterior parameter distribution. For case II, which contains biased motility and no proliferation, the only coefficients regularly identified using PDE-FIND are $c_{u_x}$ and $c_{u_{xx}}$ (table 4) and so for the ABC process we consider the PDE model

$$u_t = c_{u_{xx}} u_{xx} + c_{u_x} u_x, \tag{4.12}$$

and aim to infer the $(c_{u_{xx}}, c_{u_x})$ posterior parameter distribution. Note that PDE-FIND failed to identify one relevant model term in case II, which is $uu_x$, most likely because of either the significant noise in the samples of the dataset and/or the time scale over which the data are collected. For case III, which contains both unbiased motility and proliferation, the only coefficients that are regularly identified are $c_u$, $c_{u^2}$ and $c_{u_{xx}}$ (table 4) and so for the ABC process we consider the PDE model

$$u_t = c_u u + c_{u^2} u^2 + c_{u_{xx}} u_{xx}, \tag{4.13}$$

and aim to infer the $(c_u, c_{u^2}, c_{u_{xx}})$ posterior parameter distribution.

We explore the results of using Bayes-PDE-FIND with a form of ABC which is known as ABC-rejection sampling. At each step, $\boldsymbol{\theta}^*$ is sampled from the prior distribution $\pi(\boldsymbol{\theta})$, and the PDE given in equation (2.7) is integrated in time using parameters $\boldsymbol{\theta}^*$ to yield simulated data $\mathcal{D}_{sim}(\boldsymbol{\theta}^*)$. The simulated data, $\mathcal{D}_{sim}(\boldsymbol{\theta}^*)$, are compared with the observed data, $\mathcal{D}_{obs}$, according to a distance function $d(\mathcal{D}_{obs}, \mathcal{D}_{sim}(\boldsymbol{\theta}^*))$ provided by the practitioner. Given a *tolerance* $\varepsilon > 0$, the sampled $\boldsymbol{\theta}$ is accepted into the posterior distribution whenever $d(\mathcal{D}_{obs}, \mathcal{D}_{sim}(\boldsymbol{\theta}^*)) < \varepsilon$. We provide full details of the ABC-rejection algorithm used in this work in electronic supplementary material, §S8.

We set $\varepsilon = 0.15$ for case I, and $\varepsilon = 0.25$ for casse II and case III and run ABC rejection until a total of 300 parameters have been sampled in each of the cases. This is equivalent to an acceptance rate of approximately 15% in each of the cases. The inferred posterior distributions are shown in figures 9–11 and we refer to electronic supplementary material, §S8 for a full overview of the inferred pairwise marginal posterior distributions of the parameters in case III. In each of figures 9–11 we also show, for comparison, the posterior obtained by applying ABC rejection sampling using a broad, uniform prior on the pre-selected coefficients. For case I, we take a uniform prior where $c_{u_{xx}} \sim \mathcal{U}(0, 0.5)$ and $c_{u \cdot u_{xx}} \sim \mathcal{U}(0, 0.8)$; for case II, $c_{u_x} \sim \mathcal{U}(-0.05, 0)$ and $c_{u_{xx}} \sim \mathcal{U}(0, 0.5)$; for case III $c_u \sim \mathcal{U}(0, 0.005)$, $c_{u^2} \sim \mathcal{U}(-0.005, 0)$ and $c_{u_{xx}} \sim \mathcal{U}(0, 0.5)$. We note that the choice of uniform prior in such cases is non-trivial as it requires some prior knowledge on the part of the practitioner about the relevant parameter ranges.

### (i) Case I

In the case of unbiased motility only (case I, dataset $\mathcal{D}_I^1$; figure 9), we see that use of the spike-and-slab prior distribution (figure 9a) results in a posterior distribution with the true parameter value contained in the support of the posterior. Moreover, the sparsity enforced by the prior ensures that the correct PDE structure, with only $u_{xx}$ included, is selected. By contrast, with a uniform prior on $(c_{u_{xx}}, c_{u \cdot u_{xx}})$ (figure 9b), although the true parameter value is contained in the support of

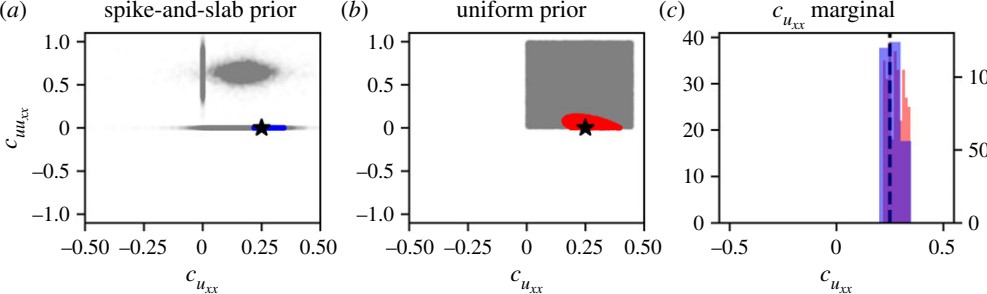

**Figure 9.** Posterior distributions obtained using dataset $\mathcal{D}_{\mathrm{I}}^1$. (*a*) Spike-and-slab joint posterior distribution (blue) together with the true parameter values (black star) and spike-and-slab prior (grey). (*b*) Joint posterior distribution, using a uniform prior (red) together with the true parameter values (black star) and uniform prior (grey). (*c*) Marginal distribution of $c_{u_{xx}}$ generated using the spike-and-slab prior (blue) and a uniform prior (red) together with the true parameter value (black dashed line). (Online version in colour.)

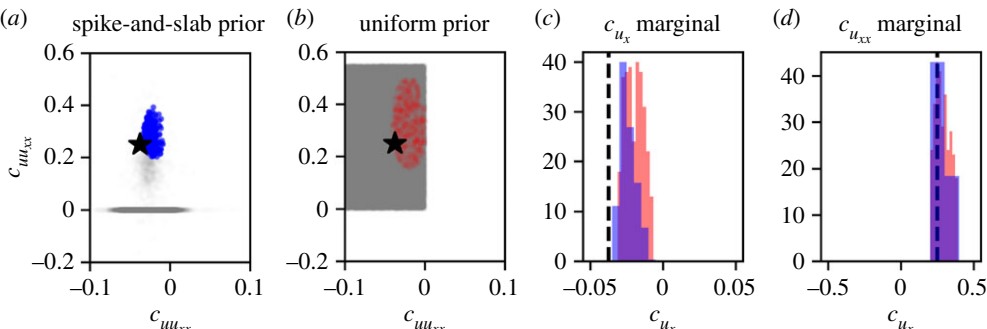

**Figure 10.** Posterior distributions obtained using dataset $\mathcal{D}_{\mathrm{II}}^1$. (*a*) Joint spike-and-slab posterior distribution (blue) together with the true parameter values (black star) and spike-and-slab prior (grey). (*b*) Joint posterior distribution, using a uniform prior distribution (red) together with the true parameter values (black star) and uniform prior (grey). (*c*) Marginal distributions of $c_{u_x}$ generated using the spike-and-slab prior (blue) and a uniform prior (red) together with the true parameter value (black dashed line). (*d*) Marginal distributions of $c_{u_{xx}}$ generated using spike-and-slab prior (blue) and a uniform prior (red) together with the true parameter value (black dashed line). (Online version in colour.)

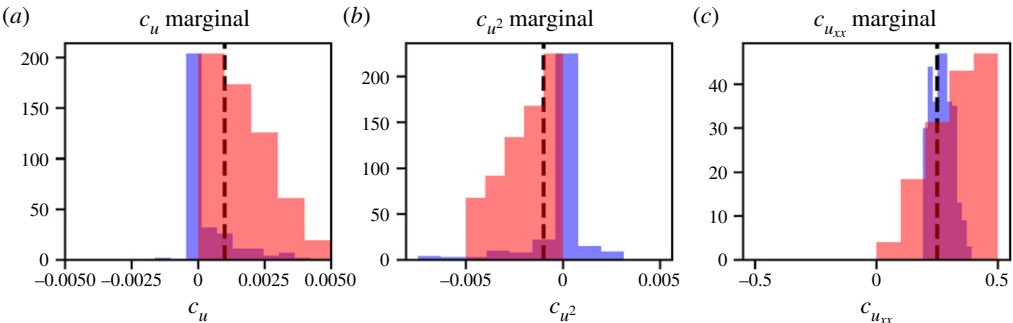

**Figure 11.** Marginal posterior distributions obtained using dataset $\mathcal{D}_{\mathrm{III}}^1$. (*a*) Marginal posterior distributions of $c_u$ generated using the spike-and-slab prior (blue) and a uniform prior (red) together with the true parameter value (black dashed line). (*b*) Marginal posterior distributions of $c_{u^2}$ generated using the spike-and-slab prior (blue) and a uniform prior (red) together with the true parameter value (black dashed line). (*c*) Marginal distributions of $c_{u_{xx}}$ generated using the spike-and-slab prior (blue) and a uniform prior (red) together with the true parameter value (black dashed line). (Online version in colour.)

the posterior distribution, the correct PDE structure is generally not established, with both $u_{xx}$ and $uu_{xx}$ terms contained in the PDE model.

### (ii) Case II

In the case of biased motility (case II, dataset $\mathcal{D}_{\text{II}}^1$; figure 10), both $c_{u_x}$ and $c_{u_{xx}}$ are non-zero for all parameter values accepted into the spike-and-slab posterior distribution. This is a striking result given an identification ratio of just 0.012 for $c_{u_{xx}}$ after the application of PDE-FIND. We remark that the posteriors in figure 10 show that in the presence of model misspecification—recall that the term $uu_x$ was not identified by PDE-FIND—the posterior distribution may be biased. In this case it entails that the true parameter values are not contained in the support of the posterior distribution.

### (iii) Case III

In the case of unbiased motility and proliferation (case III, dataset $\mathcal{D}_{\text{III}}^1$; figure 11), the posterior obtained using the PDE-FIND prior still contains some accepted parameter samples with either $c_u$ or $c_{u^2}$ equal to zero, demonstrating that there is potential non-identifiability of these terms given the data. However, all parameter samples in the posterior have non-zero $c_{u_{xx}}$, a significant result given an identification ratio of 0.57 for $c_{u_{xx}}$. We note that the support of the spike-and-slab posterior contains the true parameter value for $c_{u_{xx}}$, even though it is not in the support of the empirical distribution of the PDE-FIND coefficients from the exploration subset.

### (iv) Computational performance

To highlight the performance of our method, we compare the computational cost and the accuracy of our method against alternative options. In addition to the spike-and-slab model and uniform priors used to generate the posterior distributions (which we call *informed spike-and-slab* and *sparse uniform*), we also consider a spike-and-slab prior with mean 0 in the slab for each coefficient (i.e. the PDE-FIND screen only informs the prior through the identification ratios), which we call *naive spike-and-slab*, as well as a uniform prior on all library coefficients (i.e. the classic Bayesian scenario, where no variable selection is performed), which we refer to as *classic Bayesian*. In each of the experiments, we perform ABC rejection to sample 300 parameters from the ABC posterior with the same thresholds as used in cases I–III and record the time taken to complete. Inference is done on a Lenovo desktop computer using six Intel(R) i5-8500T cores with clock speed 2.10 GHz. Table 5 shows the time taken for each of the experiments alongside with their acceptance rates. We note that the computational time of the spike-and-slab models is significantly lower than any of the uniform prior models and that using an informed spike-and-slab prior offers a substantial speed-up in computational time compared with using a naive spike-and-slab approach. The approach using a pre-screened uniform implementation failed to yield a sparse set of coefficients, thus showing unacceptable accuracy in learning the correct equations. The classic Bayesian analysis did not finish sampling within $1.5 \times 10^6$ s (approx. two weeks). By calculating the dimensionality of the space, we estimate that the acceptance probability should be expected to be of the order of $10^{-2}$%, which confirms that performing a classical Bayesian analysis in such a case is impossible. We highlight that many applications of EQL methods will have even larger libraries, making the computational time of naive uniform priors exponentially longer. The posteriors from both naive and informed spike-and-slab models are qualitatively similar across all cases and both identify the correct regions of parameter space.

### (v) Posterior predictive check

In summary, figures 9–11 highlight that the use of PDE-FIND in combination with ABC rejection can significantly reduce the uncertainty associated with the PDE coefficients. In cases I and II, uncertainty regarding which parameter to include in the model is completely removed, as the posterior has support only on the diffusion parameter axis in case I and on a region where both

**Table 5.** Comparison of computational time and acceptance probabilities for alternatives to Bayes-PDE-FIND prior. Using an informed prior significantly outperforms all alternatives.

| method | case I | case II | case III |
|---|---|---|---|
| informed spike-and-slab | 1657 s, 24.13% | 3337 s, 16.93% | 2896 s, 19.66% |
| naive spike-and-slab | 8523 s, 6.68% | 40 867 s, 1.19% | 40 128 s, 1.08% |
| sparse uniform | 40 658 s, 1.78% | 81 975 s, 0.5% | 86 213 s, 0.5% |
| classic Bayesian | did not converge | did not converge | did not converge |

$c_{u_x}$ and $c_{u_{xx}}$ are non-zero in case II. In contrast, a uniform prior over all coefficients that have sufficiently large identification ratio (those for which $A_i > \delta$) does not enforce sparsity, which means that the resulting PDE models can be misspecified and/or contain greater complexity than is necessary to accurately predict the data. To further assess the quality of the resulting posterior parameter distributions, we carried out a posterior predictive check (figure 12). For each parameter sample accepted into the posterior distribution, we used numerical integration, as detailed in the Methods section, to obtain a prediction for the density at $t = 250$ to assess how well the model interpolates the data, and a prediction for the density at $t = 1000$, which is $t = 750$ beyond the time horizon used to train PDE-FIND on the inference subset. We then plot the 5% and 95% quantiles of the output distributions and overlay them with a representative sample from each of the datasets for cases I, II and III. The results shown in figure 12 demonstrate that the PDE model predictions can both interpolate and extrapolate the data well. We conclude that even in the presence of model misspecification, such as in case II, it is possible to obtain a posterior with reasonable predictive power, although as the time horizon is extended beyond the time horizon of the training data, the misspecification becomes apparent in the systematic prediction error. To highlight the increased accuracy, we compare this with the results shown in figure 7, where the integrated model solutions fail to resemble the empirical data when PDE-FIND is used in isolation.

# 5. Discussion and outlook

The aim of this work was to develop and showcase a framework to perform uncertainty quantification for EQL methods in the context of noisy spatio-temporal data. In essence, our approach harnesses EQL methodologies to generate an informed prior distribution, which is then used within a Bayesian framework to estimate both the model structure and posterior parameter distribution. The framework was developed in the context of the PDE-FIND EQL methodology and ABC rejection sampling, but it is sufficiently general that it could be extended and applied in the context of other EQL and Bayesian inference methodologies.

The motivation for developing such a framework stems from the fact that EQL methodologies such as PDE-FIND typically make variable predictions, in terms of both model structure and parameters, in the presence of noise, which is common for datasets in the life and biomedical sciences. Incorporating uncertainty quantification through the use of Bayesian statistics approaches provides a means to quantify the uncertainty in both model structure and parameter values, and understand how such uncertainty propagates into model predictions.

We showcased our methodology in the context of noisy spatio-temporal data generated using a canonical ABM that has seen widespread use in the modelling of motile and proliferative cell populations, and which has a corresponding PDE model that can make accurate predictions of averaged ABM output. We used datasets generated in three different parameter regimes (which incorporate different cellular behaviours) to show how to combine the advantages of the PDE-FIND algorithm (efficiency and the ability to learn simple, interpretable models) with those of ABC (ability to quantify uncertainty in parameter estimates and model predictions).

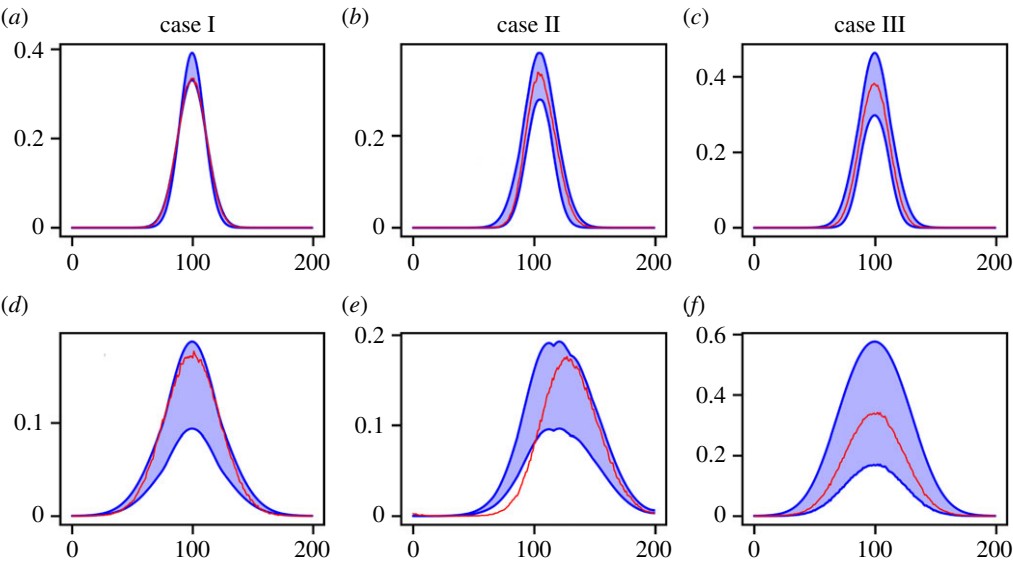

**Figure 12.** Posterior predictive check. We plot the 5% and 95% percentiles of the density distribution at each spatial location. (*a–c*) Evaluation of the interpolative capability of the PDE model using an average over all samples in the dataset (i.e. using $\langle C_i(t) \rangle_{\mathrm{LN}}$, for $i = 1, \ldots, 200$, red) at $t = 250$. (*d–f*) Evaluation of the extrapolative capability of the PDE model using an average over all samples in the dataset at $t = 1000$ (red). (Online version in colour.)

There are a number of ways in which our approach can be further improved going forward. For example, we saw that our approach may return models that fail to capture important features of the ABM, as a result of either learning the structure of the model incorrectly or inaccurate estimation of model parameter values. In the context of case II, which models biased cell motility, these inaccuracies arise partly as a result of the data used and stem from the choice of initial condition and the time scale over which the data are collected. Ultimately, the extensibility and robustness of any data-driven method are limited by the information contained in the data. Where the initial PDE-FIND screen fails to identify what are believed to be relevant terms in the PDE, it is likely that the supplied data do not provide enough information to discriminate between different forms of the PDE. In such a scenario, different experimental designs, such as a different initial condition or longer simulation time, might distinguish some of the terms of the system under consideration. We highlight that the flexibility of ABMs allows one to explore different behaviours in the model under varying experimental conditions. By analysing the effect of these variations on the resulting predictions, one can obtain important insights into how much information is contained in the data about the governing laws. This may be helpful in informing experimental design *in vitro*, so that the experimental design can provide as much information as possible. Nonetheless, the standard choice of PDE-FIND library means that it is possible for the PDE-FIND algorithm to return PDE models in which the density does not satisfy a conservation equation of the form

$$u_t = -\nabla \cdot \mathbf{F} + S(u), \tag{5.1}$$

where $\mathbf{F}$ is the flux and $S$ is the net proliferation rate, and hence for the models to make unphysical predictions (as occurs in case II). A possible solution to this specific problem could be to encode the terms in the candidate library in flux form. More generally, however, it is not obvious how to balance the wish to include constraints in specifying terms in the candidate library while at the same time avoiding over-constraining the space of possible output PDE models. Bayesian approaches may prove useful in this respect. In the case of severe model mis-specification due to incompleteness of the supplied library, Bayes-PDE-FIND offers several possibilities. In some scenarios, the learned PDE will interpolate the data well and extrapolate to new settings, even

though it contains library terms that are different from the ground truth. Such a PDE can still be used for the purposes of simulation and inference, since the benefit of such a PDE model is that it is fast to solve, which is often crucial when performing inference. When the learned PDE terms perform poorly in interpolating or extrapolating, Bayes-PDE-FIND returns a quantification of the error between model solutions and observed data. This may offer real-world insights: the library terms are usually provided by practitioners to reflect hypothesized mechanisms in the system under consideration. That those terms fail to explain the data provides a motivation to reconsider which mechanisms should form part of the model.

Secondly, our approach could be improved through the use of more efficient ABC samplers, such as ABC sequential Monte Carlo samplers that target the posterior distribution by evolving the prior distribution through a series of intermediate distributions. This requires the development of proposal distributions that can maintain the sparsity of PDE coefficients, as encoded by the PDE-FIND informed prior, so that the learned PDEs retain a simple and interpretable structure.

Data accessibility. All code to generate synthetic data, as well as code used to analyse the data, are available on Github at https://github.com/simonmape/UQ-for-pdefind.

Authors' contributions. S.M.P. created the code, produced all figures and carried out the analysis; R.E.B. and M.J.S. helped to design the study and supervised and coordinated it. S.M.P. wrote the paper, on which all other authors commented and revised. All authors gave final approval for publication.

Competing interests. We declare we have no competing interests.

Funding. S.M.P. is supported by an EPSRC/UKRI Doctoral Training Award. M.J.S. is supported by the Australian Research Council (DP200100177). R.E.B. acknowledges funding from the BBSRC via BB/R00816/1 and would like to thank the Royal Society for a Wolfson Research Merit Award.

Acknowledgements. The authors would like to thank the referees for their comments.

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
