## [Peer Review File · Proceedings. Mathematical, Physical, and Engineering Sciences]

Review History

RSPA-2021-0426.R0 (Original submission)

Review form: Referee 1

Is the manuscript an original and important contribution to its field?

Excellent

Is the paper of sufficient general interest?

Excellent

Is the overall quality of the paper suitable?

Acceptable

Can the paper be shortened without overall detriment to the main message?

Yes

Do you think some of the material would be more appropriate as an electronic appendix?

No

Do you have any ethical concerns with this paper?

No

Recommendation?

Major revision is needed (please make suggestions in comments)

Comments to the Author(s)

Minor edits included in attached document. Problem and method description is good. The exploration of the effect of noise on PDE-FIND results is interesting and novel.

Minor revisions:

Please expound on the decision to use ABC. It seems like you are in a position to derive a classical likelihood using the sample mean and covariance over your N s samples. The decision to use ABC was not well motivated as an alternative to the traditional approach.

Though not necessary, an exploration/discussion of the scenario wherein the model mis-specification was severe, e.g. an important term in the PDE was not in the dictionary passed to PDE-FIND, would help in assessing the extensibility/robustness of the approach.

For the posterior predictive check, it is my understanding that the comparison of the posterior values to data was performed for a time point that was within the time horizon of the data used in calibration. If this is true, I view this more as a check for interpolation (can the model reproduce the data it was trained on?) rather than a test for true "predictive" or extrapolative capability (can the model reproduce data not used in calibration?). I would like to see how well the posterior predictive agrees with the data beyond the time horizon used in calibration.

Major revision:

My major concern is with the definition of the prior. Using data to define the prior violates the likelihood principle, thus what is being presented is not a true Bayesian prior. Additionally, we see that in several cases the empirical prior that is being used is overly restrictive, resulting in posteriors that do not contain the true coefficients within their supports, even when the model is not mis-specified (Case III). This over-restrictiveness will lead to overconfidence in the learned model forms as well.

One of my major concerns and a source of this over-restriction is in the use of the ensemble of coefficients to define a mean and variance for the normal distribution used in the empirical prior. The form of the empirical prior reminds me of a spike and slab prior; see, e.g. Ishwaran, Hemant, and J. Sunil Rao. "Spike and slab variable selection: frequentist and Bayesian strategies." *Annals of statistics* 33.2 (2005): 730-773. DOI: 10.1214/009053604000001147. This paper discusses a range of forms of the spike and slab. I think you can use the spike and slab formulation as a basis to derive a prior that induces sparsity and doesn't rely on the data to define the support of the prior.

To summarize, I think the process of generating an ensemble of PDE-FIT solutions and using those to screen which terms are used in the Bayesian inference makes sense and I have no problem with it. My major concern is the use of the ensemble of coefficients from that process to define the prior support. I would like to see the prior redefined without this dependence on the data.

Review form: Referee 2

Is the manuscript an original and important contribution to its field?

Good

Is the paper of sufficient general interest?

Acceptable

Is the overall quality of the paper suitable?

Marginal

Can the paper be shortened without overall detriment to the main message?

Yes

Do you think some of the material would be more appropriate as an electronic appendix?

No

Do you have any ethical concerns with this paper?

No

Recommendation?

Major revision is needed (please make suggestions in comments)

Comments to the Author(s)

In this work the authors propose an equation learning strategy for agent based modelling using PDEs. Noisy data are supposed to be generated from diffusion-type equations solving several well-known PDEs with given parameters. The generated PDEs can vary in dependence of the generated sample and therefore deterministic methods are not robust in case of randomly perturbed coefficients. Hence, through a Bayesian approach they tackle the problem of parameter identification by the definition of an informative prior distribution. The problem is interesting and actual, anyway I think that several parts need to be clarified and strengthened before possible acceptance.

- 1) P.6 L. 33-38: the numerical solution of your PDE is not an issue of minimal importance. Therefore, accurate methods need to be considered. There is a huge literature on the solution of diffusion-type equations. What is the accuracy of the space discretization? What is the accuracy of time discretization? Here the authors states RK without even mentioning the order... I found this part very sloppy.
- 2) P. 8 L. 44: there is a small discussion on the norms that can be considered. Anyway a more systematic and serious comparison may be implemented. Usually L2 does not promote sparsity.
- 3) What is the computational cost of the proposed method vs its accuracy? This is not discussed but I think it is central.
- 4) Fig.10-11 are not sufficiently informative. It I would expect that the uninformative prior is centered on the expected coefficient.

Decision letter (RSPA-2021-0426.R0)

03-Aug-2021

Dear Mr Martina Perez

The Editor of Proceedings A has now received comments from referees on the above paper and would like you to revise it in accordance with their suggestions which can be found below (not including confidential reports to the Editor).

Please submit a copy of your revised paper within four weeks - if we do not hear from you within this time then it will be assumed that the paper has been withdrawn. In exceptional circumstances, extensions may be possible if agreed with the Editorial Office in advance.

Please note that it is the editorial policy of Proceedings A to offer authors one round of revision in which to address changes requested by referees. If the revisions are not considered satisfactory by the Editor, then the paper will be rejected, and not considered further for publication by the

journal. In the event that the author chooses not to address a referee's comments, and no scientific justification is included in their cover letter for this omission, it is at the discretion of the Editor whether to continue considering the manuscript.

To revise your manuscript, log into <http://mc.manuscriptcentral.com/prsa> and enter your Author Centre, where you will find your manuscript title listed under "Manuscripts with Decisions." Under "Actions," click on "Create a Revision." Your manuscript number has been appended to denote a revision.

You will be unable to make your revisions on the originally submitted version of the manuscript. Instead, revise your manuscript and upload a new version through your Author Centre.

When submitting your revised manuscript, you will be able to respond to the comments made by the referee(s) and upload a file "Response to Referees" in Step 1: "View and Respond to Decision Letter". Please provide a point-by-point response to the comments raised by the reviewers and the editor(s). A thorough response to these points will help us to assess your revision quickly. You can also upload a 'tracked changes' version either as part of the 'Response to reviews' or as a 'Main document'.

IMPORTANT: Your original files are available to you when you upload your revised manuscript. Please delete any unnecessary previous files before uploading your revised version.

When revising your paper please ensure that it remains under 28 pages long. In addition, any pages over 20 will be subject to a charge (£150 + VAT (where applicable) per page). Your paper has been ESTIMATED to be 25 pages.

Open Access

You are invited to opt for open access, our author pays publishing model. Payment of open access fees will enable your article to be made freely available via the Royal Society website as soon as it is ready for publication. For more information about open access please visit <https://royalsociety.org/journals/authors/open-access/>. The open access fee for this journal is £1700/\$2380/€2040 per article. VAT will be charged where applicable. Please note that if the corresponding author is at an institution that is part of a Read and Publishing deal you are required to select this option. See <https://royalsociety.org/journals/librarians/purchasing/read-and-publish/read-publish-agreements/> for further details.

Once again, thank you for submitting your manuscript to Proc. R. Soc. A and I look forward to receiving your revision. If you have any questions at all, please do not hesitate to get in touch.

Yours sincerely
Raminder Shergill
proceedingsa@royalsociety.org

on behalf of
Professor Vincenzo Capasso
Board Member
Proceedings A

Reviewer(s)' Comments to Author:
Referee: 1

Comments to the Author(s)
Minor edits included in attached document. Problem and method description is good. The exploration of the effect of noise on PDE-FIND results is interesting and novel.

Minor revisions:

Please expound on the decision to use ABC. It seems like you are in a position to derive a classical likelihood using the sample mean and covariance over your N_s samples. The decision to use ABC was not well motivated as an alternative to the traditional approach.

Though not necessary, an exploration/discussion of the scenario wherein the model mis-specification was severe, e.g. an important term in the PDE was not in the dictionary passed to PDE-FIND, would help in assessing the extensibility/robustness of the approach.

For the posterior predictive check, it is my understanding that the comparison of the posterior values to data was performed for a time point that was within the time horizon of the data used in calibration. If this is true, I view this more as a check for interpolation (can the model reproduce the data it was trained on?) rather than a test for true "predictive" or extrapolative capability (can the model reproduce data not used in calibration?). I would like to see how well the posterior predictive agrees with the data beyond the time horizon used in calibration.

Major revision:

My major concern is with the definition of the prior. Using data to define the prior violates the likelihood principle, thus what is being presented is not a true Bayesian prior. Additionally, we see that in several cases the empirical prior that is being used is overly restrictive, resulting in posteriors that do not contain the true coefficients within their supports, even when the model is not mis-specified (Case III). This over-restrictiveness will lead to overconfidence in the learned model forms as well.

One of my major concerns and a source of this over-restriction is in the use of the ensemble of coefficients to define a mean and variance for the normal distribution used in the empirical prior. The form of the empirical prior reminds me of a spike and slab prior; see, e.g. Ishwaran, Hemant, and J. Sunil Rao. "Spike and slab variable selection: frequentist and Bayesian strategies." *Annals of statistics* 33.2 (2005): 730-773. DOI: 10.1214/009053604000001147. This paper discusses a range of forms of the spike and slab. I think you can use the spike and slab formulation as a basis to derive a prior that induces sparsity and doesn't rely on the data to define the support of the prior.

To summarize, I think the process of generating an ensemble of PDE-FIT solutions and using those to screen which terms are used in the Bayesian inference makes sense and I have no problem with it. My major concern is the use of the ensemble of coefficients from that process to define the prior support. I would like to see the prior redefined without this dependence on the data.

Referee: 2

Comments to the Author(s)

In this work the authors propose an equation learning strategy for agent based modelling using PDEs. Noisy data are supposed to be generated from diffusion-type equations solving several well-known PDEs with given parameters. The generated PDEs can vary in dependence of the generated sample and therefore deterministic methods are not robust in case of randomly perturbed coefficients. Hence, through a Bayesian approach they tackle the problem of parameter identification by the definition of an informative prior distribution. The problem is interesting and actual, anyway I think that several parts need to be clarified and strengthened before possible acceptance.

1) P.6 L. 33-38: the numerical solution of your PDE is not an issue of minimal importance.

Therefore, accurate methods need to be considered. There is a huge literature on the solution of diffusion-type equations. What is the accuracy of the space discretization? What is the accuracy of time discretization? Here the authors states RK without even mentioning the order... I found this part very sloppy.

2) P. 8 L. 44: there is a small discussion on the norms that can be considered. Anyway a more systematic and serious comparison may be implemented. Usually L2 does not promote sparsity.

- 3) What is the computational cost of the proposed method vs its accuracy? This is not discussed but I think it is central.
- 4) Fig.10-11 are not sufficiently informative. It I would expect that the uninformative prior is centered on the expected coefficient.

Board Member:

Comments to Author(s):

Based on the comments of both Reviewers the paper requires major revisions.

Please include a Response file in which all comments have been addressed one by one, in the style QA (Question-Answer).

The application was unable to attach manuscript files to this email, because one or more of the files exceeded the allowable attachment size (6MB).

Author's Response to Decision Letter for (RSPA-2021-0426.R0)

See Appendix A.

RSPA-2021-0426.R1 (Revision)

Review form: Referee 1

Is the manuscript an original and important contribution to its field?

Excellent

Is the paper of sufficient general interest?

Excellent

Is the overall quality of the paper suitable?

Excellent

Can the paper be shortened without overall detriment to the main message?

Yes

Do you think some of the material would be more appropriate as an electronic appendix?

No

Do you have any ethical concerns with this paper?

No

Recommendation?

Accept with minor revision (please list in comments)

Comments to the Author(s)

I am happy with the changes made to the manuscript and methods. I appreciate the additional discussion re. the need for ABC and the updated prior specification, as well as extrapolation of the posterior predictive well beyond the calibration time horizon. I noticed a couple minor

changes that may be necessary, listed below.

Manuscript:

Algorithm 2 does not seem to have been updated to reflect the new prior specification. Figure S15, the marginals don't seem consistent with those presented in Figure 11, e.g. `cuxx` support contains the true value in Figure 11 but not in S15--are they supposed to be the same case? This plot may be a holdover from the previous prior specification and may need to be updated.

Source code:

It does not appear that the provided source code has been updated to reflect the new prior specification. Please update the github repository.

Review form: Referee 2

Is the manuscript an original and important contribution to its field?

Good

Is the paper of sufficient general interest?

Good

Is the overall quality of the paper suitable?

Good

Can the paper be shortened without overall detriment to the main message?

Yes

Do you have any ethical concerns with this paper?

No

Recommendation?

Accept as is

Comments to the Author(s)

The Authors answered to my concerns and I suggest acceptance of the paper in its present form.

Decision letter (RSPA-2021-0426.R1)

24-Sep-2021

Dear Mr Martina Perez,

On behalf of the Editor, I am pleased to inform you that your Manuscript RSPA-2021-0426.R1 entitled "Bayesian uncertainty quantification for data-driven equation learning" has been accepted for publication subject to minor revisions in Proceedings A. Please find the referees' comments below.

The reviewer(s) have recommended publication, but also suggest some minor revisions to your manuscript. Therefore, I invite you to respond to the reviewer(s)' comments and revise your manuscript. Please note that we have a strict upper limit of 28 pages for each paper. Please endeavour to incorporate any revisions while keeping the paper within journal limits. Please

note that page charges are made on all papers longer than 20 pages. If you cannot pay these charges you must reduce your paper to 20 pages before submitting your revision. Your paper has been ESTIMATED to be 27 pages. We cannot proceed with typesetting your paper without your agreement to meet page charges in full should the paper exceed 20 pages when typeset. If you have any questions, please do get in touch.

It is a condition of publication that you submit the revised version of your manuscript within 7 days. If you do not think you will be able to meet this date please let me know in advance of the due date.

To revise your manuscript, log into <https://mc.manuscriptcentral.com/prsa> and enter your Author Centre, where you will find your manuscript title listed under "Manuscripts with Decisions." Under "Actions," click on "Create a Revision." Your manuscript number has been appended to denote a revision.

You will be unable to make your revisions on the originally submitted version of the manuscript. Instead, revise your manuscript and upload a new version through your Author Centre.

When submitting your revised manuscript, you will be able to respond to the comments made by the referee(s) and upload a file "Response to Referees" in Step 1: "View and Respond to Decision Letter". Please provide a point-by-point response to the comments raised by the reviewers and the editor(s). A thorough response to these points will help us to assess your revision quickly. You can also upload a 'tracked changes' version either as part of the 'Response to reviews' or as a 'Main document'.

IMPORTANT: Your original files are available to you when you upload your revised manuscript. Please delete any redundant files before completing the submission process.

When uploading your revised files, please make sure that you include the following as we cannot proceed without these:

- 1) A text file of the manuscript (doc, txt, rtf or tex), including the references, tables (including captions) and figure captions. Please remove any tracked changes from the text before submission. PDF files are not an accepted format for the "Main Document".
- 2) A separate electronic file of each figure (tif, eps or print-quality pdf preferred). The format should be produced directly from original creation package, or original software format.
- 3) Electronic Supplementary Material (ESM): all supplementary materials accompanying an accepted article will be treated as in their final form. Note that the Royal Society will not edit or typeset supplementary material and it will be hosted as provided. Please ensure that the supplementary material includes the paper details where possible (authors, article title, journal name). Supplementary files will be published alongside the paper on the journal website and posted on the online figshare repository (<https://figshare.com>). The heading and legend provided for each supplementary file during the submission process will be used to create the figshare page, so please ensure these are accurate and informative so that your files can be found in searches. Files on figshare will be made available approximately one week before the accompanying article so that the supplementary material can be attributed a unique DOI. Alternatively you may upload a zip folder containing all source files for your manuscript as described above with a PDF as your "Main Document". This should be the full paper as it appears when compiled from the individual files supplied in the zip folder.

Article Funder

Please ensure you fill in the Article Funder question on page 2 to ensure the correct data is collected for FundRef (<http://www.crossref.org/fundref/>).

Media summary

Please ensure you include a short non-technical summary (up to 100 words) of the key findings/importance of your paper. This will be used for to promote your work and marketing purposes (e.g. press releases). The summary should be prepared using the following guidelines:

*Write simple English: this is intended for the general public. Please explain any essential technical terms in a short and simple manner.

*Describe (a) the study (b) its key findings and (c) its implications.

*State why this work is newsworthy, be concise and do not overstate (true 'breakthroughs' are a rarity).

*Ensure that you include valid contact details for the lead author (institutional address, email address, telephone number).

Cover images

We welcome submissions of images for possible use on the cover of Proceedings A. Images should be square in dimension and please ensure that you obtain all relevant copyright permissions before submitting the image to us. If you would like to submit an image for consideration please send your image to proceedingsa@royalsociety.org

Open Access

You are invited to opt for open access, our author pays publishing model. Payment of open access fees will enable your article to be made freely available via the Royal Society website as soon as it is ready for publication. For more information about open access please visit <https://royalsociety.org/journals/authors/open-access/>. The open access fee for this journal is £1700/\$2380/€2040 per article. VAT will be charged where applicable. Please note that if the corresponding author is at an institution that is part of a Read and Publishing deal you are required to select this option. See <https://royalsociety.org/journals/librarians/purchasing/read-and-publish/read-publish-agreements/> for further details.

Once again, thank you for submitting your manuscript to Proceedings A and I look forward to receiving your revision. If you have any questions at all, please do not hesitate to get in touch.

Best wishes

Raminder Shergill

proceedingsa@royalsociety.org

Proceedings A

on behalf of

Professor Vincenzo Capasso

Board Member

Proceedings A

Reviewer(s)' Comments to Author:

Referee: 1

Comments to the Author(s)

I am happy with the changes made to the manuscript and methods. I appreciate the additional discussion re. the need for ABC and the updated prior specification, as well as extrapolation of the posterior predictive well beyond the calibration time horizon. I noticed a couple minor changes that may be necessary, listed below.

Manuscript:

Algorithm 2 does not seem to have been updated to reflect the new prior specification.

Figure S15, the marginals don't seem consistent with those presented in Figure 11, e.g. `cuxx` support contains the true value in Figure 11 but not in S15--are they supposed to be the same case? This plot may be a holdover from the previous prior specification and may need to be updated.

Source code:

It does not appear that the provided source code has been updated to reflect the new prior specification. Please update the github repository.

Referee: 2

Comments to the Author(s)

The Authors answered to my concerns and I suggest acceptance of the paper in its present form.

Board Member

Comments to Author(s):

Based on the comments by the Reviewers, the Authors are welcome to submit a revised version taking them into account.

Decision letter (RSPA-2021-0426.R2)

30-Sep-2021

Dear Mr Martina Perez

I am pleased to inform you that your manuscript entitled "Bayesian uncertainty quantification for data-driven equation learning" has been accepted in its final form for publication in Proceedings A.

Our Production Office will be in contact with you in due course. You can expect to receive a proof of your article soon. Please contact the office to let us know if you are likely to be away from e-mail in the near future. If you do not notify us and comments are not received within 5 days of sending the proof, we may publish the paper as it stands.

As a reminder, you have provided the following 'Data accessibility statement' (if applicable).

Statement (if applicable): All code to generate synthetic data, as well as code used to analyse the data is available on Github at <https://github.com/simonmape/UQ-for-pdefind>.

Open access

You are invited to opt for open access, our author pays publishing model. Payment of open access fees will enable your article to be made freely available via the Royal Society website as soon as it is ready for publication. For more information about open access please visit <https://royalsociety.org/journals/authors/which-journal/open-access/>. The open access fee for this journal is £1700/\$2380/€2040 per article. VAT will be charged where applicable.

Note that if you have opted for open access then payment will be required before the article is published – payment instructions will follow shortly.

If you wish to opt for open access then please inform the editorial office (proceedingsa@royalsociety.org) as soon as possible.

Your article has been estimated as being 27 pages long. Our Production Office will inform you of the exact length at the proof stage.

Proceedings A levies charges for articles which exceed 20 printed pages. (based upon approximately 540 words or 2 figures per page). Articles exceeding this limit will incur page charges of £150 per page or part page, plus VAT (where applicable).

Under the terms of our licence to publish you may post the author generated postprint (ie. your accepted version not the final typeset version) of your manuscript at any time and this can be made freely available. Postprints can be deposited on a personal or institutional website, or a recognised server/repository. Please note however, that the reporting of postprints is subject to a media embargo, and that the status the manuscript should be made clear. Upon publication of the definitive version on the publisher's site, full details and a link should be added.

You can cite the article in advance of publication using its DOI. The DOI will take the form: 10.1098/rspa.XXXX.YYYY, where XXXX and YYYY are the last 8 digits of your manuscript number (eg. if your manuscript number is RSPA-2017-1234 the DOI would be 10.1098/rspa.2017.1234).

For tips on promoting your accepted paper see our blog post:
<https://royalsociety.org/blog/2020/07/promoting-your-latest-paper-and-tracking-your-results/>

On behalf of the Editor of Proceedings A, we look forward to your continued contributions to the Journal.

Sincerely,
Raminder Shergill
proceedingsa@royalsociety.org

on behalf of
Professor Vincenzo Capasso
Board Member
Proceedings A

Appendix A

Mathematical Institute
Andrew Wiles Building, Radcliffe Observatory Quarter
Woodstock Road, Oxford, OX2 6GG

Professor Michael Lockwood FRS
Editor-in-chief
Proceedings of the Royal Society A

27th August 2021

Dear Professor Lockwood,

Re-submission of manuscript number RSPA-2021-0426

I am writing to submit the revised version of our manuscript “Bayesian uncertainty quantification for data-driven equation learning” for consideration in *Proceedings of the Royal Society A*.

We would like to take this opportunity to thank the reviewers for their constructive comments. In our revised manuscript, we have carefully addressed all comments, with changes in the revised manuscript marked in **purple**.

We hope that you will now find our manuscript suitable for publication. Please do not hesitate to get in touch if you have any queries.

Best wishes,

Simon Martina-Perez
Doctoral candidate in mathematics
University of Oxford

Bayesian uncertainty quantification for data-driven equation learning

S. Martina-Perez, M. J. Simpson and R. E. Baker

Response to comments by Reviewer 1

Minor edits included in attached document. Problem and method description is good. The exploration of the effect of noise on PDE-FIND results is interesting and novel.

We are glad that Reviewer 1 is supportive of publication of our manuscript. We would like to thank them for their constructive comments, which we respond to in full below.

1. Minor edits in the attached document

We agree with the reviewer on all comments regarding the text, and have made all suggested changes accordingly.

2. Please expound on the decision to use ABC. It seems like you are in a position to derive a classical likelihood using the sample mean and covariance over your N_s samples. The decision to use ABC was not well motivated as an alternative to the traditional approach.

We agree with the reviewer that more detail can be provided for our decision to use a likelihood-free method. In section 4a we have now added a paragraph to elucidate why the likelihood function is not mathematically tractable, and so classical likelihoods cannot be used. We have also added a detailed mathematical discussion of the issue (copied below) to Section S7 of the supplementary information.

The likelihood $P(\mathcal{D}_{\text{obs}} | \theta)$ defines the probability density of the observations \mathcal{D}_{obs} given the model parameters θ . In this context, the observed data $\{U(x, t)\}$ at space points $x = x_1, x_2, \dots, x_N$ and time points $t = t_1, t_2, \dots, t_N$ are obtained from a stochastic ABM. The solution $u(x, t; \theta)$ of the PDE model is an approximation of the mean of the ABM data, *i.e.*:

$$u(x, t; \theta) \approx \mathbb{E}_{\theta}[U(x, t)]. \quad (1)$$

To define a classical likelihood, one would need to first assume that the mean of the ABM data is exactly given by the PDE solution and describe the distribution of ABM outputs around the PDE model mean by

$$U(x_i, t_i) = u(x_i, t_i; \theta) + \epsilon_i, \quad (2)$$

where ϵ_i is a random variable with zero mean (not necessarily identically distributed, but generally assumed independent). One can then obtain a likelihood for the observed data by prescribing a probability distribution for the ϵ_i , which would yield a likelihood for $U(x, t)$ given $u(x, t; \theta)$. However, for a general ABM, the distribution for the deviation from the mean (*i.e.* the distribution of the ϵ_i) is unknown. In some cases, one might choose to make a simplifying assumption on the ϵ_i , such as a Gaussian approximation. This may be appropriate when one is familiar with the noise process and the distribution of the data is approximately Gaussian, for example in the case of a very large number of samples, where the Central Limit Theorem can be invoked. However, in the small data limit considered in EQL applications, the assumption that the ϵ_i are Gaussian is unreasonable. It will depend on the details of the ABM as to the extent to which realisations vary from their mean, which is for the purposes of inference, unknown. As we prefer to avoid placing unnecessary assumptions on the process, in the form of assumptions on the ϵ_i , we opt instead for a likelihood-free approach. We additionally stress that classical likelihood models, when considering a likelihood for general PDE data, employ the assumption that the errors ϵ_i are independent in space and time, and so the joint likelihood of the data given the model,

$$P(U(x_1, t_1), \dots, U(x_N, t_N) | u(x_1, t_1; \theta), \dots, u(x_N, t_N; \theta)), \quad (3)$$

can be decomposed as

$$P(U(x_1, t_1), \dots, U(x_N, t_N) | u(x_1, t_1; \theta), \dots, u(x_N, t_N; \theta)) = \prod_i P(U(x_i, t_i) | u(x_i, t_i; \theta)). \quad (4)$$

Since individual realisations of the ABM may have deviations from the mean that have spatial and temporal structure, it is inappropriate to assume that the ϵ_i are independent. To summarise,

a classical likelihood approach can be used, but it requires the introduction of nontrivial additional assumptions and so it will potentially not target the true posterior. We prefer not to take this approach, and rather opt for a method that has been developed specifically to deal with cases where the likelihood is mathematically intractable.

3. Though not necessary, an exploration/discussion of the scenario wherein the model mis-specification was severe, e.g. an important term in the PDE was not in the dictionary passed to PDE-FIND, would help in assessing the extensibility/robustness of the approach.

We agree with the reviewer that an explicit exploration/discussion of the scenario of model mis-specification would help in assessing the extensibility and robustness of the approach. We have added the text below to the discussion section in the manuscript.

Ultimately, the extensibility and robustness of any data-driven method are limited by the information contained in the data. Where the initial PDE-FIND screen fails to identify what are believed to be relevant terms in the PDE, it is likely that the supplied data does not provide enough information to discriminate between different forms of the PDE. In such a scenario, different experimental designs, such as a different initial condition or longer simulation time, might distinguish some of the terms of the system under consideration. We highlight that the flexibility of ABMs allows one to explore different behaviours in the model under varying experimental conditions. By analysing the effect of these variations on the resulting predictions, one can obtain important insights into how much information is contained in the data about the governing laws. This may be helpful in informing experiment design *in vitro*, so that the experimental design can provide as much information as possible.

In the case of severe model mis-specification due to incompleteness of the supplied library, Bayes-PDE-FIND offers several possibilities. In some scenarios, the learned PDE will interpolate the data well and extrapolate to new settings, even though it contains library terms that are different from the ground truth. Such a PDE can still be used for the purposes of simulation and inference, since the benefit of such a PDE model is that it is fast to solve, which is often crucial when performing inference. When the learned PDE terms perform poorly in interpolating or extrapolating, Bayes-PDEFIND returns a quantification of the error between model solutions and observed data. This may offer real-world insights: the library terms are usually provided by practitioners to reflect hypothesised mechanisms in the system under consideration. That those terms fail to explain the data provides a motivation to reconsider which mechanisms should form part of the model.

4. For the posterior predictive check, it is my understanding that the comparison of the posterior values to data was performed for a time point that was within the time horizon of the data used in calibration. If this is true, I view this more as a check for interpolation (can the model reproduce the data it was trained on?) rather than a test for true "predictive" or extrapolative capability (can the model reproduce data not used in calibration?). I would like to see how well the posterior predictive agrees with the data beyond the time horizon used in calibration.

We fully agree with the reviewer on this point. To provide analysis of the model to predict unseen data, i.e. to extrapolate to a longer time horizon, we have added a figure where we use the posterior distribution over model parameters to predict the solution at a much later time point, and generated independent data traces from the ABM to display alongside this. The updated version of Figure 12 shows that on a substantially longer time horizon (i.e. $t = 1000$ compared to calibration time horizon $t = 500$), the posterior distribution over parameters still produces realistic predictions. As expected, the resulting uncertainty bounds in all cases are wider, reflecting the variability in the learned posteriors. We conclude that even in the presence of model misspecification, such as in Case II, it is possible to obtain a posterior with reasonable predictive power, although as the time horizon is extended beyond the time horizon of the calibration data, the misspecification becomes apparent in the systematic prediction error.

5. My major concern is with the definition of the prior. Using data to define the prior violates the likelihood principle, thus what is being presented is not a true Bayesian prior. Additionally, we see that in several cases the empirical prior that is being used is overly restrictive, resulting in posteriors that do not contain the true coefficients within their supports, even when the model is not mis-specified (Case III). This over-restrictiveness will lead to overconfidence in the learned model forms as well. One of my major concerns and a source of this over-restriction is in the use of the ensemble of coefficients to define a mean and variance for the normal distribution used in the empirical prior. The form of the empirical prior reminds me of a spike and slab prior; see, e.g.

Ishwaran, Hemant, and J. Sunil Rao. "Spike and slab variable selection: frequentist and Bayesian strategies." *Annals of statistics* 33.2 (2005): 730-773. DOI: 10.1214/009053604000001147. This paper discusses a range of forms of the spike and slab. I think you can use the spike and slab formulation as a basis to derive a prior that induces sparsity and doesn't rely on the data to define the support of the prior. To summarize, I think the process of generating an ensemble of PDE-FIT solutions and using those to screen which terms are used in the Bayesian inference makes sense and I have no problem with it. My major concern is the use of the ensemble of coefficients from that process to define the prior support. I would like to see the prior redefined without this dependence on the data.

We thank the reviewer for this observation and for making us aware of the possibility to perform inference with a spike-and-slab prior. We have repeated our analysis using the slab-and-spike prior and included the results in the revised manuscript. We have added the text below to Section 4 of the manuscript.

Spike-and-slab models are powerful tools to perform variable selection in regression problems [1, 2, 3, 4]. The main idea of a spike-and-slab type prior is that it defines a two-point mixture distribution in which coefficients are mutually independent. Each mixture is made up of a flat distribution with large support (the slab) and a degenerate distribution at zero (the spike). In early formulations, the slab was modeled as a uniform distribution over some region of parameter space [3, 4], whereas in more recent work, inference is performed on hyperparameters of the marginal distributions [1, 2]. Samples of the hyperparameters yielding a high variance will lead to sampling parameters far away from zero, whereas samples of the hyperparameters yielding a low variance will sample close to zero. In this way, the aim is to explore parameter space by iteratively sampling over the hyperparameters and the values for the coefficients using Gibbs sampling. In this work, we wish to exploit the simplicity of the earliest slab-and-spike models, which use a Dirac measure at zero to enforce sparsity, whilst utilising as much information as possible from the PDE-FIND screen in defining the prior distribution without violating the likelihood principle. We follow the hierarchical Bayesian group LASSO model with an independent spike and slab type prior for each coefficient [2]. We set the group size in the model of Xu et al. [2] equal to one, so that the prior $\pi_i(\xi_i)$ for each coefficient ξ_i is given by

$$\begin{aligned}\xi_i | \mu, \sigma_i^2 &\sim (1 - a_i)\delta_0 + a_i\mathcal{N}(\mu_i, \sigma_i^2), \\ \sigma_i^2 &\sim \mathcal{IG}(\alpha_i, \beta_i),\end{aligned}$$

where \mathcal{IG} is the inverse-gamma distribution with parameters α_i, β_i that define the shape of the prior on σ_i^2 . This is the standard choice for modeling the distribution of the hypervariances. In the approach of Xu et al. [2], $\mu = 0$. In this work, we can use knowledge of the coefficients gained through our initial PDE-FIND screen to inform the μ_i . To do so, we randomly divide the ABM data in half and use one subset in the PDE-FIND screen to inform the prior (exploration subset), and the other subset to perform inference (inference subset). We first set a_i equal to the i -th identification ratio and μ_i equal to the i -th sample mean of the PDE-FIND coefficients trained on the exploration subset. Since the hyperprior for the variances σ_i^2 allows for large values of σ_i^2 , this prior is not overly restrictive, since values far away from the sample mean can be sampled. Second, we tune α_i, β_i using the exploration subset so that the variance is on average the same order of magnitude as the PDE-FIND coefficients. This is crucial: a large (small) variance in parameters that are typically small (large) will fail to sample from the relevant regions of parameter space. The exact values of μ_i, α_i, β_i are given in Supplementary Information Section S9. These considerations now imply that the prior for ξ is given by

$$\boldsymbol{\pi} = \bigotimes_{i=1}^{N_\ell} \left\{ \mathbb{I}(i \in A)\pi_i\xi_i + \mathbb{I}(i \notin A) \cdot \delta_0 \right\}. \quad (5)$$

We make two additional observations to answer the reviewer's comment. The first is that our approach now respects the likelihood principle: all the information of the sample used in the inference step of Bayes-PDEFIND enters the inference process only through the approximate likelihood function since it is distinct from the sample that was used to inform the global properties of the prior. In addition, we note that the findings for the new spike-and-slab model are qualitatively very similar to the findings with the marginals defined earlier by the PDE-FIND data. The main difference is that none of the priors are overly restrictive: the true parameter is contained within the support of

each of the priors. We have also explored (see our answer also to question 3 of Reviewer 2) the case where $\mu = 0$, as is common in some works in the literature. While the results are qualitatively identical to the approach where the mean of the slab is informed by a prior screen, we note that the computational cost is increased by an order of magnitude (in our inference typically by a factor 10).

Response to comments by Reviewer 2

In this work the authors propose an equation learning strategy for agent based modelling using PDEs. Noisy data are supposed to be generated from diffusion-type equations solving several well-known PDEs with given parameters. The generated PDEs can vary in dependence of the generated sample and therefore deterministic methods are not robust in case of randomly perturbed coefficients. Hence, through a Bayesian approach they tackle the problem of parameter identification by the definition of an informative prior distribution. The problem is interesting and actual, anyway I think that several parts need to be clarified and strengthened before possible acceptance.

1. P.6 L. 33-38: the numerical solution of your PDE is not an issue of minimal importance. Therefore, accurate methods need to be considered. There is a huge literature on the solution of diffusion-type equations. What is the accuracy of the space discretization? What is the accuracy of time discretization? Here the authors states RK without even mentioning the order... I found this part very sloppy.

We have added details about the space and time discretization, as well as the order of the Runge-Kutta method to the text on page 6.

The resulting ODEs are solved using a fourth-order Runge-Kutta method on the domain $x \in [0, 200]$ with space discretisation $\Delta x = 10^{-3}$ and constant time discretisation $\Delta t = 10^{-4}$.

2. P. 8 L. 44: there is a small discussion on the norms that can be considered. Anyway a more systematic and serious comparison may be implemented. Usually L2 does not promote sparsity.

We agree with the reviewer that the choice of norm is relevant. However, as we note in our discussion, the specific choice of the norm has been an active part of the research into equation learning (EQL), and there is still no definitive answer as to which norm should be preferred. The choice of the L^2 norm is, as we note in the discussion and referenced papers, the standard implementation in the PDE-FIND algorithm. The scope of this work is to investigate uncertainty when implementing the PDE-FIND algorithm. While it certainly is interesting to investigate how the algorithm may be improved by a different algorithmic design (in this case, by using a different norm), this falls outside the scope of this work. We would like to note our detailed discussion of the sequential thresholding procedure within PDE-FIND to enforce sparsity. While the L^2 norm alone does not enforce sparsity, the design of the algorithm is geared to combine the strengths of L^2 regularization with sparsity. This is evidenced by observing the sparsity of PDE-FIND predictions we obtain: only a few parameters have a nonzero identification ratio in our experiments.

3. What is the computational cost of the proposed method vs its accuracy? This is not discussed but I think it is central.

We thank the reviewer for this comment. We have performed a systematic comparison of the computational cost of our new spike-and-slab approach with three available alternatives. We have added the following discussion to the results section of the manuscript.

*To highlight the performance of our method, we compare the computational cost and the accuracy of our method against alternative options. In addition to the spike-and-slab model and uniform priors used to generate the posterior distributions (which we call *informed spike-and-slab* and *sparse uniform*, we also consider a spike-and-slab prior with mean 0 in the slab for each coefficient (i.e. the PDE-FIND screen only informs the prior through the identification ratios), which we call *naive spike-and-slab*, as well as a uniform prior on all library coefficients (i.e. the classic Bayesian scenario, where no variable selection is performed), which we refer to as *classic Bayesian*. In each of the experiments, we perform ABC rejection to sample 300 parameters from the ABC posterior with the same thresholds as used in Case I-III and record the time taken to complete. Inference is done on a Lenovo desktop computer using 6 Intel(R) i5-8500T cores with clock speed 2.10GHz. The table below reports the computational time for each of the methods alongside the acceptance rates in each of the ABC rejection implementations.*

Method	Case I	Case II	Case III
Informed spike-and-slab	1657 s , 24.13%	3337 s , 16.93%	2896 s , 19.66%
Naive spike-and-slab	8523 s , 6.68%	40867 s , 1.19%	40128 s , 1.08%
Sparse uniform	40658 s , 1.78%	81975 s , 0.5%	86213 s , 0.5%
Classic Bayesian	Did not converge	Did not converge	Did not converge

We note that the computational time of the spike-and-slab models is significantly lower than any of the uniform prior models and that using an informed spike-and-slab prior offers a substantial speed-up in computational time than using a naive spike-and-slab approach. The approach using a pre-screened uniform implementation failed to yield a sparse set of coefficients, thus showing unacceptable accuracy in learning the correct equations. The classic Bayesian analysis did not finish sampling within $1.5 \cdot 10^6 s$ (approximately two weeks). By calculating the dimensionality of the space, we estimate that the acceptance probability should be expected to be of order $10^{-2}\%$, which confirms that performing a classical Bayesian analysis in such a case is inappropriate. We highlight that many applications of EQL methods will have even larger libraries, making the computational time of naive uniform priors exponentially longer. The posteriors from both naive and informed spike-and-slab models are qualitatively similar across all cases and both identify the correct regions of parameter space.

- Fig.10-11 are not sufficiently informative. It would be expected that the uninformative prior is centered on the expected coefficient.

This work considers the problem where a partial differential equation (PDE) is to be learned from data in cases where the practitioner has no information about the structure of the PDE. The goal is to find a sparse set of coefficients so that the PDE can be expressed as a linear combination of terms in a pre-defined library. If one has access to a large amount of detailed and uncorrupted data, then we agree with the reviewer that centering the priors around the expected values of the coefficients is a sensible choice. However, in many applications where EQL is used, one cannot count on having such data. Therefore, in a practical (experimental) scenario, robust methods that do not make strong assumptions about the PDE terms are needed. In our experiments, we have observed that model mis-specification can occur as a consequence of the data being insufficient, incomplete or noisy. In Case II, for instance, this mis-specification resulted in the learned PDE not containing one term that should have been contained in the PDE. However, a posterior distribution away from the expected coefficient still performed well in predicting the data. Thus, it is possible that models with parameters away from the expected values adequately capture the data and it is crucial that the prior distribution allows to sample such regions in parameter space.

In the case of Figures 10 and 11, this means that the prior must indeed be defined on a very large region of parameter space and absolutely not around the values that we earlier identified as defining the ground truth PDE. The point of our having chosen this ABM and PDE is that we can do EQL and then test the predictions against what we know are the true PDE coefficients. This knowledge of the PDE, however, cannot enter the definition of the uninformed prior: the point of Figures 10-11 is to showcase the difference in performance between choosing a method that simplifies searching through parameter space against a method that assumes no knowledge whatsoever and needs to explore all possible values.

References

- [1] Hemant Ishwaran and J. Sunil Rao. Spike and slab variable selection: Frequentist and Bayesian strategies. *The Annals of Statistics*, 33(2):730 – 773, 2005.
- [2] Xiaofan Xu and Malay Ghosh. Bayesian variable selection and estimation for group lasso. *Bayesian Analysis*, 10(4), Dec 2015.
- [3] T. J. Mitchell and J. J. Beauchamp. Bayesian variable selection in linear regression. *Journal of the American Statistical Association*, 83(404):1023–1032, 1988.
- [4] T. Kloek and F. B. Lempers. Posterior Probabilities Of Alternative Linear Models. *Econometric Institute Archives*, (272033), February 1970.